# COFlowNet: Conservative Constraints on Flows Enable High-Quality Candidate Generation

**Yudong Zhang**[1], **Xuan Yu**[1], **Xu Wang**[1,2,*],
**Zhaoyang Sun**[1], **Chen Zhang**[1], **Pengkun Wang**[1,2], **Yang Wang**[1,2,3] [*]
1. University of Science and Technology of China (USTC), Hefei, China
2. Suzhou Institute for Advanced Research, USTC, Suzhou, China
3. State Key Laboratory of Precision and Intelligent Chemistry, USTC, Hefei, China
{zyd2020@mail., yx2024@mail., wx309@}ustc.edu.cn
{sunzhaoyang@mail., zhangchenzc@mail., pengkun@, angyan@}ustc.edu.cn

## Abstract

Generative flow networks (GFlowNet) have been considered as powerful tools for generating candidates with desired properties. Given that evaluating the property of candidates can be complex and time-consuming, existing GFlowNets train proxy models for efficient online evaluation. However, the performance of proxy models is heavily dependent on the amount of data and is of considerable uncertainty. Therefore, it is of great interest that how to develop an offline GFlowNet that does not rely on online evaluation. Under the offline setting, the limited data results in an insufficient exploration of state space. The insufficient exploration means that offline GFlowNets can hardly generate satisfying candidates out of the distribution of training data. Therefore, it is critical to restrict the offline model to act in the distribution of training data. The distinctive training goal of GFlownets poses a unique challenge for making such restrictions. Tackling the challenge, we propose Conservative Offline GFlowNet (COFlowNet) in this paper. We define unsupported flow, edges containing unseen states in training data. Models can learn extremely little knowledge about unsupported flow from training data. By constraining the model from exploring unsupported flows, we restrict COFlowNet to explore as optimal trajectories on the training set as possible, thus generating better candidates. In order to improve the diversity of candidates, we further introduce a quantile version of unsupported flow restriction. Experimental results on several widely-used datasets validate the effectiveness of COFlowNet in generating high-scored and diverse candidates. All implementations are available at https://github.com/yuxuan9982/COflownet.

## 1 Introduction

Reinforcement learning (RL) is typically about finding an optimal solution to a given target (Mnih et al., 2015; Sutton, 2018). RL models are required to generate the single highest-reward sequence of actions. However, it has become increasingly apparent that the ability to produce a variety of candidate solutions, not just the optimal one, is highly valuable for numerous real-world applications, including molecule design (Huang et al., 2016; Zhang et al., 2021; Bengio et al., 2021) and exploration in RL (Hazan et al., 2019). For example, in the scenario of molecule design, rather than generating a high-scoring molecule that cannot be synthesized, the model should generate a series of molecules with suboptimal scores, so that chemists can pick molecules that are easier to synthesize.

Generative Flow Networks (GFlowNets) (Bengio et al., 2021; Jain et al., 2022; Bengio et al., 2023; Gao et al., 2022) have emerged as a potent tool for generating diverse candidates. The key insight of GFlowNets is to ensure that the probability of generating a candidate is proportional to the positive reward associated with that candidate. Therefore, GFlowNets is able to sample a series of possible

---

[*]Dr. Xu Wang and Prof. Yang Wang are the corresponding authors. Email: wx309@ustc.edu.cn

candidates with the reward distribution. Taking advantage of such ability, GFlowNets have expressed promising potential in many object generation application areas. Jain et al. (2022) embeds GFlowNet into an active learning framework for biological sequence design, which iteratively generates diverse candidates and screens the candidates to enhance the training of GFlowNet. Deleu et al. (2022); Nishikawa-Toomey et al. (2022) leverage GFlowNets as a general framework for generative modeling of discrete and composite objects, which approximates the posterior distribution over the structure of Bayesian networks. Liu et al. (2023a) leverages GFlowNets for sampling structured sub-network modules, thereby enhancing predictive robustness. Zhang et al. (2023a;c) apply GFlowNets to address combinatorial optimization challenges. Additionally, Zhou et al. (2023) explores the use of GFlowNets for phylogenetic inference, demonstrating the model's versatility in diverse applications.

In many scenarios, evaluating generated candidates could be expensive and time-consuming, making it impractical to calculate the accurate score of candidates. For example, in the realm of drug design, evaluating a potential molecule often requires conducting biological experiments or performing complex chemical calculations. Given that these assessment procedures can span from several minutes to multiple days, they are impractical to integrate directly into the training phase of an RL model. To this point, existing GFlowNets follow the method proposed in Angermueller et al. (2019), which suggests training a proxy model based on evaluated candidates to approximate the accurate scores (rewards) of candidates. Specifically, we have a set of candidates $X = \{x_i\}$ and a set of corresponding scores $Y = \{y_i | y_i = oracal(x_i)\}$, where $oracal$ denotes the expensive but accurate evaluation of candidates. Based on the dataset $(X, Y)$, a proxy model $f : x \rightarrow y$ can be trained to approximate $oracal$. The proxy model is then employed to calculate the rewards of online sampled candidates, with which the GFlowNets can be trained.

While GFlowNets have achieved notable success across various domains of object generation, their effectiveness is significantly contingent upon the quality of the proxy models they rely on. Typically, GFlowNets are trained to align with the candidate-reward distribution as estimated by a proxy model. However, the scarcity of data can introduce considerable variability in the proxy model's accuracy. If the proxy model fails to accurately mirror the true quality of the candidates, the resulting performance of the GFlowNets will be suboptimal. Given that proxy model training requires a comprehensive dataset of candidates along with their actual scores, an alternative approach involves the development of offline reinforcement learning methods. These methods could potentially sidestep the pitfalls associated with proxy model dependency, offering a more reliable framework for GFlowNets training.

Unlike conventional RL models, GFlowNets are trained with the specific objective of generating candidates with probabilities that are directly proportional to the positive rewards linked to those candidates. This distinctive training goal presents unique challenges for the development of offline GFlowNets, making existing offline RL techniques can not be directly applied. Specifically, the policy in GFlowNet frameworks is determined by inflows and outflows of states, which complicates the application of actor-critic methods (Nair et al., 2020; Tarasov et al., 2024). In Q-learning frameworks (Kostrikov et al., 2021), Q-values are maximized iteratively by the Bellman operation, which is actually against the flow match constraint in GFlowNet. Policy constraint or matching methods (Wang et al., 2023) may assimilate less desirable state space, such as some low-reward areas, making them unsuitable for offline GFlowNets. In essence, the distinctive training goal of GFlowNets calls for the development of new offline techniques not found in traditional RL approaches.

In this paper, we propose a novel offline training strategy for GFlowNets to make full utilization of collected data, called Conservative Offline GFlowNet (COFlowNet). To avoid the generation of highly uncertain candidates, we define unsupported flows and propose to regularize the unsupported flows, so that the model can learn informative knowledge from training data. To enhance the diversity of generated candidates, we introduce a quantile matching algorithm, and modify the regularization of unsupported flows into quantile style. We evaluate the proposed offline training strategy following the experimental setting of Bengio et al. (2021). By applying the proposed training objective, the offline version of GFlowNet shows great potential for generating diverse and high-score candidates.

The main contributions of this paper are as follows:

- Our research endeavors to adapt GFlowNets for offline scenarios. A pioneering offline training strategy named Conservative Offline GFlowNet (COFlowNet) is introduced. Central to our approach is the concept of unsupported flows. By regularizing unsupported flows, the model learns informative knowledge from training data and generates high-score candidates.

- To enhance the diversity of generated candidates, we introduce a quantile matching algorithm. By modifying the regularization of unsupported flows into quantile style, we achieve the final training objective of COFlowNet, termed conservative quantile matching (CQM).

- We evaluate the proposed offline training strategy in alignment with the experimental setting in Bengio et al. (2021). The proposed COFlowNet, equipped with the novel offline training objective, exhibits significant promise in producing a spectrum of diverse and high-performing candidates.

## 2 RELATED WORK

**GFlowNet.** Since the introduction of GFlowNets by Bengio et al. (2021), there has been a surge of research in this domain, covering various aspects of the technology. Malkin et al. (2022); Zimmermann et al. (2022) have explored the relationship with variational methods, demonstrating that GFlowNets surpass variational inference when utilizing off-policy training data. Pan et al. (2022; 2023b) has established frameworks to enhance credit assignment efficiency by incorporating intermediate signals within GFlowNets. Jain et al. (2022) has delved into multi-objective generation capabilities, while Pan et al. (2023c) integrated world modeling. An unsupervised learning approach for GFlowNets was suggested by Pan et al. (2023a), and Ma et al. (2024) examined the use of isomorphism tests to mitigate flow bias in training. From a probabilistic modeling perspective, Zhang et al. (2022b) has concurrently trained an energy-based model alongside a GFlowNet, validating its effectiveness on discrete data modeling tasks and proposing a bidirectional proposal mechanism later adopted by Kim et al. (2023) for local search algorithms. Zhang et al. (2022a); Lahlou et al. (2023); Zhang et al. (2023b) provided a theoretical analysis and bridged the gap between diffusion modeling and GFlowNets. GFlowNets have also shown promise in numerous object generation applications, including biological sequence design by Jain et al. (2022) and causal structure learning by Deleu et al. (2022). Considering the complexity and time intensity associated with candidate property evaluation, current GFlowNets utilize proxy models to facilitate efficient online assessments. Yet, these proxy models' effectiveness is highly contingent upon data volume, introducing a significant margin of uncertainty. Hence, there is considerable interest in developing an offline GFlowNet that is independent of real-time evaluation mechanisms. However, GFlowNets train to generate candidates with probabilities proportional to their rewards, presenting challenges for offline adaptation that don't align with standard offline RL techniques. In the following, we analyze why existing offline RL methods are not applicable.

**Offline RL.** Most existing offline RL methods are based on an actor-critic framework or Q-learning framework. In a flow-based framework, a policy $\pi$ is directly given by $\pi(a|s) = F(s,a)/F(s)$, where $s$ denotes a state with actions $a \in \mathcal{A}(s)$ and $F(s,a)$ denotes the flow of state $s$ taking action $a$. The policy is too fixed, making it difficult to apply techniques of actors from actor-critic-based methods of offline reinforcement learning, such as AWAC (Nair et al., 2020) and ReBRAC (Tarasov et al., 2024). However, Q-learning-based frameworks are also not applicable. The concept of flows is analogous to Q-values in Q-learning and the critic in the actor-critic method. However, there are still some significant differences between them. Q-values are iterated by the Bellman operation, estimating how good an action is. But they focus on the maximum value, while the values of flows denote the total sum of rewards passing the state, making it impractical to apply methods such as Kostrikov et al. (2021) to our flow matching objective. For the flow matching objective to learn the behavior policy of offline datasets better, an intuitive way is to use policy constraint methods (Wang et al., 2023). Policy constraint methods will force the behavior strategy to learn the bad parts of the dataset, such as some low-reward areas, thus it is not ideal for offline GFlowNets. Our proposed method avoids this problem by not directly forcing trained policy to stay close to behavior policy.

## 3 METHOD

### 3.1 PROBLEM FORMULATION

We describe the problem of interest here. We aim at training a policy to generate candidate objects $x \in \mathcal{X}$ with probability proportional to a reward function $R(x) : x \to \mathbb{R}^+$. We generate a candidate object $x$ from an initial state $s_0$, and make a series of actions to transfer the state finally into $x$. The procedure can be described by a trajectory of state transformation, denoted as $\tau = (s_0, s_1, s_2, \cdots, s_n = x)$.

We denote the set of states as $\mathcal{S}$, and the set of actions as $\mathcal{A} = \{(s \to s')|s, s' \in \mathcal{S}\}$. Note that we here assume the relationship between action and future state is a one-to-one correspondence, *i.e.*, there is only one action $(s \to s')$ that transfers $s$ to $s'$. We say $s$ is a parent of $s'$ and $s'$ is a child of $s$ when we have $(s \to s') \in \mathcal{A}$. Specially, we have $\mathcal{A}(s)$ denoting the set of actions between $s$ and all its children, and we thus have $\mathcal{A}(x) = \varnothing$ for any terminal states $x$.

Generative Flow Networks (Bengio et al., 2021) (GFlowNets) are developed for the target that generates candidate objects $x$ with probability proportional to $R(x)$. Such an objective is achieved in GFlowNets by casting the set of action trajectories as a flow and converting the flow consistency equations into a learning objective.

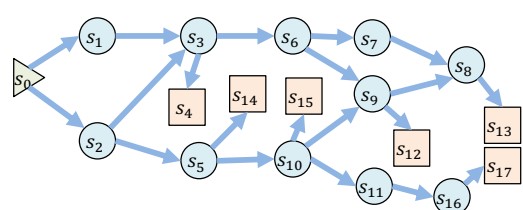

As in Fig. 1, the state transformation can be illustrated as a directed acyclic graph (DAG). In the flow network, each node represents a state, and each edge represents a flow (action and its corresponding probability). The source (or root) node only generates outflows. Interior nodes have both inflows and outflows, with inflows equal to outflows. Leaf nodes (or terminal states) only receive inflows and store them as sinks. At

Figure 1: **Illustration of the flow network.** The triangle means initial state, the circles denote interior states, and the squares denote the terminal states. At each interior state, we have its inflow and outflow matched. For a terminal state, the inflow of it is equal to the reward at it.

a specific state $s_i$, we have several input flows from its prior states and output flows to its successive states. The flow consistency equations require the inflow and outflow of an interior state (node) to be matched, and the inflow of a terminal state is the reward of the state. We define $F(s, s') : (s, s') \to \mathbb{R}^+$ as flow between state $s$ and $s'$. By setting $R(s) = 0$ for interior states and $\mathcal{A}(s) = \varnothing$ for terminal states, the flow match constraint of state $s$ is given as,

$$\mathcal{L}_{FM}(s) = \sum_{s:(s \to s') \in \mathcal{A}} F(s, s') - R(s') - \sum_{s'':(s' \to s'') \in \mathcal{A}} F(s', s'') \tag{1}$$

The above equation constrains that the inflows of a state are equal to the outflows of the state plus the reward of the state. Considering a whole trajectory $\tau$, we have,

$$\mathcal{L}_{FM}(\tau) = \sum_{s \in \tau \neq s_0} \mathcal{L}_{FM}^2(s) \tag{2}$$

Note that we square $\mathcal{L}_{FM}(s)$ to ensure the value is positive. The training objective of GFlowNets is to learn $F(s, s')$ to minimize $\mathcal{L}_{FM}(\tau)$. Bengio et al. (2021) proves that a global optimum of the expected loss provides the correct flows. And the training objective can be achieved by setting the probability of action $(s \to s')$ as,

$$P((s \to s')|s) = \frac{F(s, s')}{\sum_{s'':(s \to s'') \in \mathcal{A}(s)} F(s, s'')} \tag{3}$$

This equation suggests that the probability of taking action $(s \to s')$ is the ratio of the flow of this action to the outflow of $s$.

## 3.2 CONSERVATIVE GFLOWNET FOR OFFLINE RL

Due to the innovative design of GFlowNets, *i.e.*, the flow matching, existing offline RL frameworks are not applicable. To this end, this paper proposes a conservative offline GFlowNet (COFlowNet), which learns informative knowledge from training data and shows the ability to generate diverse candidates.

In the offline setting, the model is constrained from extensively exploring the state space and must rely solely on the provided training data. This limitation can lead to inadequate coverage of the state space within the data. Consequently, the performance of candidates whose trajectories include many states not present in the training data may be highly uncertain, as limited clues of their performance can be learned from training data.

To avoid the generation of highly uncertain candidates, COFlowNet makes constraints on the flows. The offline dataset $\mathcal{D}$ we used is composed by a list of transitions $(s_t, a_t, s_{t+1}, r_t, d_t)_i$ , where $i$ indexes a transition sampled from a trajectory $\tau$. Specifically, we call a flow $(s \rightarrow s')$ supported if there exists a trajectory $(s_0 \rightarrow s_1 \rightarrow s_2 \rightarrow \cdots \rightarrow s_n)$ in the dataset $\mathcal{D}$ such that $s_t = s$ and $s_{t+1} = s'$. Otherwise, it is unsupported. These supported flows compose our action set $\mathcal{A}_{\mathcal{D}}$ of training data and serve as the basis for imposing node-specific constraints within the proposed COFlowNet framework.

Figure 2: **Illustration of supported and unsupported flows.** Edges not present in the dataset are defined as unsupported flows. For instance, $(s_2 \rightarrow s_3)$ is unsupported.

For inflows in state $s$, we constrain the unsupported inflows in the dataset by adding a regularization term to constrain them into small values. The unseen actions are thus constrained to help COFlowNet better learn the behavior policy of training data. The regularization term of unsupported inflows of state $s$ can be formulated as,

$$\mathcal{R}_{in}(s) = \sum_{s':(s' \rightarrow s) \in \mathcal{A}} F(s', s) - \sum_{s'_{\mathcal{D}}:(s'_{\mathcal{D}} \rightarrow s) \in \mathcal{A}_{\mathcal{D}}} F(s'_{\mathcal{D}}, s) \tag{4}$$

For outflows in node $s \in \mathcal{S}$, our strategy is similar, and the regularization term of outflows is defined as,

$$\mathcal{R}_{out}(s) = \sum_{s'':(s \rightarrow s'') \in \mathcal{A}} F(s, s'') - \sum_{s''_{\mathcal{D}}:(s \rightarrow s''_{\mathcal{D}}) \in \mathcal{A}_{\mathcal{D}}} F(s, s''_{\mathcal{D}}) \tag{5}$$

Let us define two trade-off factors $\alpha_1 \geq 0$ and $\alpha_2 \geq 0$ for $\mathcal{R}_{in}(s)$ and $\mathcal{R}_{out}(s)$. We can turn our Equation into our constrained flow matching (CFM) objective for interior and terminal states to optimize the parameter $\theta$.

$$\mathcal{L}_{CFM}(s) = \mathcal{L}^2_{FM}(s) + \alpha_1 \mathcal{R}^2_{in}(s) + \alpha_2 \mathcal{R}^2_{out}(s) \tag{6}$$

We here square $\mathcal{R}_{in}$ and $\mathcal{R}_{out}$ for the same reason as Eq. 2. We next prove that applying the regularization will exactly decrease the unsupported flow and will not hurt the supported flow.

**Theorem 1.** Given two flow estimation function, $\hat{F}$ trained with regularization and $F$ trained without regularization, we have $\hat{F}(s, s') \leq F(s, s')$ obtains for unsupported flow $(s, s')$, *i.e.*, $(s, s') \in \mathcal{A}$ and $(s, s') \notin \mathcal{A}_{\mathcal{D}}$.

**Proof**. With trainable parameters in $F$ denoted as $\theta$, the derivative of constrained flow matching objective to $\theta$ is,

$$\frac{\partial \hat{\mathcal{L}}_{CFM}(s)}{\partial \theta} = 2\mathcal{L}_{FM}(s) \cdot \left( \sum_{s':(s' \rightarrow s) \in \mathcal{A}} \frac{\partial \hat{F}(s', s)}{\partial \theta} - \sum_{s'':(s \rightarrow s'') \in \mathcal{A}} \frac{\partial \hat{F}(s, s'')}{\partial \theta} \right)$$

$$+ 2\alpha_1 \mathcal{R}_{in}(s) \cdot \left( \sum_{s':(s' \rightarrow s) \in \mathcal{A}} \frac{\partial \hat{F}(s', s)}{\partial \theta} - \sum_{s_{\mathcal{D}}:(s_{\mathcal{D}} \rightarrow s) \in \mathcal{A}_{\mathcal{D}}} \frac{\partial \hat{F}(s_{\mathcal{D}}, s)}{\partial \theta} \right) \tag{7}$$

$$+ 2\alpha_2 \mathcal{R}_{out}(s) \cdot \left( \sum_{s'':(s \rightarrow s'') \in \mathcal{A}} \frac{\partial \hat{F}(s, s'')}{\partial \theta} - \sum_{s''_{\mathcal{D}}:(s \rightarrow s''_{\mathcal{D}}) \in \mathcal{A}_{\mathcal{D}}} \frac{\partial \hat{F}(s, s''_{\mathcal{D}})}{\partial \theta} \right)$$

And we have the derivative of flow matching objective without regularization as,

$$\frac{\partial \mathcal{L}_{FM}(s)}{\partial \theta} = 2\mathcal{L}_{FM}(s) \cdot \left( \sum_{s':(s' \rightarrow s) \in \mathcal{A}} \frac{\partial F(s', s)}{\partial \theta} - \sum_{s'':(s \rightarrow s'') \in \mathcal{A}} \frac{\partial F(s, s'')}{\partial \theta} \right) \tag{8}$$

The second and third terms of the derivative in Eq. 7 only minimize flows not appear in the offline dataset. We get $F(\hat{s}, s') = F(s, s')$ only when $\alpha_1 = 0$ and $\alpha_2 = 0$.

**Theorem 2.** Given any state, minimizing its inflow from unsupported states will enlarge the inflow from supported states.

**Proof.** For any state $s$, for example, $s_4$ in Figure 3, we have that the total inflows equal the total outflows:

$$F(s_1, s_4) + F(s_2, s_4) + F(s_3, s_4) = F(s_4, S_c)$$

where $S_c$ denotes all the children of $s_4$ and $F(s_4, S_c)$ is the sum of outflows from $s_4$ to its children. The total reward (flow) in the training data is determined, as all terminal states are given by the data, and the reward can only be obtained at terminal states. Therefore, the outflow $F(s_4, S_c)$ is constant and denoted as $F_o$. Furthermore, the inflow from supported states $F(s_1, s_4) + F(s_2, s_4)$ is equal to $F_o - F(s_3, s_4)$. Since $F_o$ is constant, minimizing the inflow from unsupported states $F(s_3, s_4)$ will lead to an increase in the inflow from supported states $F(s_1, s_4) + F(s_2, s_4)$. This reasoning process can be easily extended to general conditions (with more supported and unsupported states).

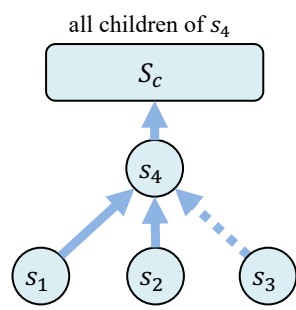

Figure 3: **Illustration of Theorem 2.** Due to the balance of inflows and outflows, reducing the flow between $s_3$ and $s_4$ will increase the other two inflows.

Directly using Eq. 6 will result in large flow at states near to $s_0$. To solve the numerical issue, we proposed the log sum exp form of our objective similar to Bengio et al. (2021),

$$\mathcal{L}_{CFM}(s) = (\log[\epsilon + \sum_{s':(s' \to s) \in \mathcal{A}} \exp(F_\theta^{\log}(s', s))] - \log[\epsilon + R(s) + \sum_{s'':(s \to s'') \in \mathcal{A}} \exp(F_\theta^{\log}(s, s''))])^2$$

$$+ \alpha_1 (\log[\epsilon + \sum_{s':(s' \to s) \in \mathcal{A}} \exp(F_\theta^{\log}(s', s))] - \log[\epsilon + \sum_{s'_\mathcal{D}:(s'_\mathcal{D} \to s) \in \mathcal{A}_\mathcal{D}} \exp(F_\theta^{\log}(s'_\mathcal{D}, s))])^2$$

$$+ \alpha_2 (\log[\epsilon + \sum_{s'':(s \to s'') \in \mathcal{A}} \exp(F_\theta^{\log}(s, s''))] - \log[\epsilon + \sum_{s''_\mathcal{D}:(s \to s''_\mathcal{D}) \in \mathcal{A}_\mathcal{D}} \exp(F_\theta^{\log}(s, s''_\mathcal{D}))])^2 \tag{9}$$

### 3.3 BETTER DIVERSITY WITH QUANTILE MATCHING FLOWS

We here introduce and modify the quantile matching algorithm to consider the uncertainty of reward and thus enhance the diversity of generated candidates of our framework. Quantile matching algorithm is originally used to handle situations where the reward function is stochastic and outperforms deterministic flow matching algorithms even on deterministic datasets (Zhang et al., 2023d).

Follow the definition in Zhang et al. (2023d), we use $Z_\beta(s, s')$ to represent the quantile flow between $s$ and $s'$. And the equality of inflows and outflows is expanded to the distribution between two random variables, *i.e.*, the quantile flow,

$$\delta^{\beta, \hat{\beta}}(s) = \log \sum_{(s' \to s) \in \mathcal{A}} \exp(Z_\beta^{\log}(s', s)) - \log \sum_{(s \to s'') \in \mathcal{A}} \exp(Z_{\hat{\beta}}^{\log}(s, s'')) \tag{10}$$

We thus have the regularization term of unsupported inflows in Eq. 4 as,

$$\delta_{in}^\beta(s) = \log \sum_{(s' \to s) \in \mathcal{A}} \exp Z_\beta^{\log}(s', s) - \log \sum_{(s'_\mathcal{D} \to s) \in \mathcal{A}_\mathcal{D}} \exp Z_\beta^{\log}(s'_\mathcal{D}, s) \tag{11}$$

Similarly, the regularization term of unsupported outflows of $s$ is formed as this,

$$\delta_{out}^{\hat{\beta}}(s) = \log \sum_{(s \to s'') \in \mathcal{A}} \exp Z_{\hat{\beta}}^{\log}(s, s'') - \log \sum_{(s \to s''_\mathcal{D}) \in \mathcal{A}_\mathcal{D}} \exp Z_{\hat{\beta}}^{\log}(s, s''_\mathcal{D}) \tag{12}$$

We deploy the pinball error $\rho_\beta(\delta) = |\beta - \mathbf{1}\{\delta < 0\}|\ell_1(\delta)$ to both $\delta_{in}^\beta$ and $\delta_{out}^{\hat{\beta}}$, where $\ell_1(\cdot)$ is a smooth $\ell_1$ loss:

$$\ell_1(\delta) = \begin{cases} \frac{1}{2}\delta^2 & \text{if } |\delta| < 1 \\ |\delta| - \frac{1}{2} & \text{otherwise} \end{cases} \tag{13}$$

Finally, we propose the conservative quantile matching (CQM) objective for COFlowNet,

$$\mathcal{L}_{CQM}(s) = \frac{1}{\hat{N}} \sum_{i=1}^{N} \sum_{j=1}^{\hat{N}} \rho_{\beta_i}(\delta^{\beta_i, \hat{\beta}_j}(s)) + \alpha_1 \frac{1}{N} (\sum_{i=1}^{N} \rho_{\beta_i}(\delta_{in}^{\beta_i}(s))) + \alpha_2 \frac{1}{\hat{N}} (\sum_{i=1}^{\hat{N}} \rho_{\hat{\beta}_i}(\delta_{out}^{\hat{\beta}_i}(s))) \quad (14)$$

## 4 EXPERIMENT

In this section, we evaluate the proposed COFlowNet on two tasks, Hypergrid and molecule design. During the evaluation, we mainly focus on two research questions: 1) How is the performance of candidates generated by COFlowNet? 2) How is the diversity of the generated candidates? To facilitate a more comprehensive evaluation, we select various metrics tailored to different tasks. These metrics are chosen to better evaluate the *performance* and *diversity*. Besides the two questions, we will also investigate the impact of different components of COFlowNet. Additionally, we deploy the proposed COFlowNet to additional tasks in B. All the experiments are conducted on an NVIDIA Tesla A100 80GB. The offline dataset is formatted as $\mathcal{D} = \{s, s', r | (s, s') \in \mathcal{A}_{\mathcal{D}}, r = R(s')\}$. When training the model, we sample batched data from $\mathcal{D}$ and calculate the loss function. We can thus align the samples used for training in our offline model with states visited in online models for fair comparison. We detail the experimental settings and results on specific tasks in the following.

### 4.1 HYPERGRID

#### 4.1.1 TASK DEFINITION

We first evaluate the proposed COFlowNet on the hypergrid task from Bengio et al. (2021). The state space is a $D$-dimensional hypercube grid of size $H^D$, where $H$ represents the dimension of the grid. The agent is tasked with formulating long-term plans and learning from sparse reward signals. It begins at the origin of the grid, *i.e.*, at coordinate $(0, 0, \cdots, 0)$, and must navigate by incrementing one of the coordinates by 1 with each move. Additionally, the agent has the option to execute a special termination action from any state. Upon deciding to stop, the agent is awarded a reward as specified by the following reward function,

$$R(x) = R_0 + R_1 \prod_{d=1}^{D} \mathbb{I}(|\frac{x_d}{H-1} - 0.5| \in (0.25, 0.5])) + R_2 \prod_{d=1}^{D} \mathbb{I}(|\frac{x_d}{H-1} - 0.5| \in (0.3, 0.4]))$$

$$(15)$$

Where $\mathbb{I}$ is the indicator function, which returns 1 when the input condition is true otherwise 0, we set $R_0 = 0.00001$, $R_1 = 0.5$, $R_2 = 2$, $H = 8$ and $D = 4$. The formula reveals that there are $2^D = 16$ distinct modes for this task, where a mode is defined as a local region (potentially encompassing one or more states) that yields the highest reward value.

#### 4.1.2 OFFLINE DATA CONSTRUCTION

Three strategies are applied to construct the offline dataset for training COFlowNet on Hypergrid: 1) **Expert:** employ an online GFlowNet to generate an offline dataset with $2 \times 10^4$ trajectories. 2) **Random:** randomly generate an offline dataset with $2 \times 10^4$ trajectories. 3) **Mixed:** take $10^4$ trajectories from **Expert** and $10^4$ trajectories from **Random**.

#### 4.1.3 RESULT

We report two metrics of COFlowNet, *number of modes* and $\ell_1$ *error*. *Number of modes* denotes how many modes the model finds, which reflects the diversity of the model. As mentioned, the goal of GFlowNets is to generate candidates with probabilities proportional to their rewards. In this case, we can enumerate all states and accurately give the ground truth the probability that a candidate is generated as $p(x) = R(x)/\sum_{x \in \mathcal{X}} R(x)$. And we can approximate the probability of generating $x$, denoted as $\pi(x)$, by repeated sampling and counting frequencies for $x$. $\ell_1$ *error* is defined as $\mathbb{E}[|p(x) - \pi(x)|]$, which estimates whether the model generates candidates with probabilities proportional to their rewards.

This task is rather simple, we employ vanilla GFlowNet (Bengio et al., 2021) for comparison. Since the optimal diversity of candidates is rather small, we apply CFM loss in Eq. 9 on this task rather

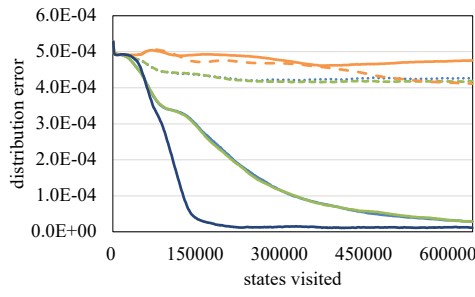 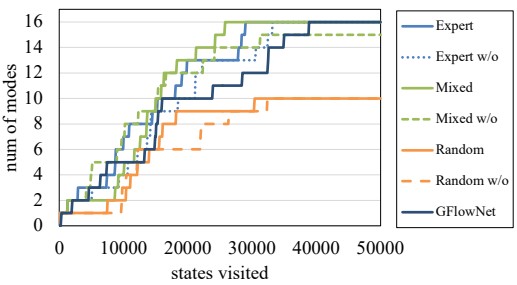

(a) $\ell_1$ error

(b) number of modes

Figure 4: **Experimental result on Hypergrid. Expert**, **Random** and **Mixed** correspond to different settings of constructing offline datasets as described in Sec. 4.1.2. *w/o* means removing the regularization term from the training loss.

than CQM loss in Eq. 14. The result on Hypergrid is reported in Fig. 4, where *w/o* means removing regularization term from the training loss, *i.e.*, using only flow matching loss and setting $\alpha_i$ to 0.

As reported in Fig. 4b, most models find all of the 16 modes within 40000 times state visiting. When trained on Mixed data, COFlowNet can generate all the modes with the least visit of states. Applying the proposed regularization method will reduce the number of state visits by 10%. It is worth noting that the model performs badly on the Random dataset no matter regularization is applied or not. Such a result indicates that the performance of offline models relies on the quality of training data.

Fig. 4a reports the $\ell_1$ error of different models. We can find that vanilla GFlowNet achieves the goal of generating candidates with probabilities proportional to their rewards with the least times of state visiting. It can be found that when trained on Expert and Mixed, COFlowNet can also get close to the goal of GFlowNets, but takes more times of state visiting. Actually, due to the limited coverage of state space of training data, COFlowNet can hardly really achieve the goal. It is also worth noting that the $\ell_1$ error is high when the regularization is removed. The reason can be found in Eq. 7. When the regularization is removed, the model can not find any clue to reduce the unsupported flow, resulting in randomly exploring those flows, which is against the goal. For example, suppose that we have a supported flow $(s_1, x)$ and an unsupported flow $(s_2, x)$ flow into the same terminal state $x$. Then, the reward of $x$ is divided randomly into the two flows (only related to the initialization of $F$). Without the regularization, the model will never know whether the flow should be divided into $(s_2, x)$. Worse still, the model even needs to allocate inflow for the state $s_2$ to balance the inflow and outflow of $s_2$.

Overall, through the result of this simple task, we find the potential of COFlowNet and the effectiveness of the proposed conservative regularization strategy. Next, we introduce a more complex and realistic task: molecule design.

## 4.2 MOLECULE DESIGN

### 4.2.1 TASK DEFINITION

In recent years, artificial intelligence methods have flourished in the field of chemistry, especially for molecular-related tasks (Du et al., 2024; Li et al., 2024; Zhang et al., 2023e). Among those tasks, molecule design is a typical application scenario for GFlowNets, where both the diversity and quality of candidates are required. In this task, the objective is to design a variety of molecules that exhibit specific chemical properties. The emphasis on diversity in this task is crucial, as it allows chemists to discover molecules that are not only characterized by their desired chemical properties but are also easy to synthesize. In this section, we are interested in designing molecules with large binding energy to a particular protein target.

In this task, we have states as molecule graphs or SMILES [1], and actions as adding new components to a molecule. Therefore, the molecule design task becomes a decision process. We have a vocabulary

---

[1] https://en.wikipedia.org/wiki/Simplified_Molecular_Input_Line_Entry_System

| Models | Data scarcity | | | Fully trained | | |
|---|---|---|---|---|---|---|
| | avg top 10 | avg top 100 | avg top 1000 | avg top 10 | avg top 100 | avg top 1000 |
| MARS | / | / | / | 8.0778 | 7.833 | 7.5992 |
| PPO | / | / | / | 8.4249 | 8.3387 | 8.2555 |
| GFlowNet | 8.4381 | 8.2909 | 8.0580 | 8.5283 | 8.3539 | 8.1440 |
| QM-GFlowNet | 8.4979 | 8.3272 | 8.1014 | 8.5552 | 8.4019 | 8.1886 |
| COFlowNet w/o | 8.3400 | 8.2046 | 8.0996 | 8.4859 | 8.3278 | 8.1083 |
| COFlowNet | 8.4638 | 8.3034 | 8.1088 | 8.5029 | 8.3730 | 8.1693 |

Table 1: **The average reward of the top $k$ candidates.** Darker blue denotes the best result and lighter blue denotes the second best.

of building blocks specified by junction tree modeling (Jin et al., 2018), which we inherit from vanilla GFlowNet (Bengio et al., 2021). At each step, the action space is determined by two factors: selecting an atom to which a building block will be attached, and deciding which block to attach. In our case, the size of the vocabulary of building blocks is 105. Given a molecule, a building block can be added to the molecule at different positions. And the number of applicable actions of a state is greater than 105, leading to a larger state space.

### 4.2.2 DATASET

As computing binding energy is computationally expensive, existing online RL models train proxy models to approximate it. Interestingly, the proxy models are also employed to evaluate their molecule models for computational convenience. If the target of an RL model is to fit the distribution of reward function, there is nothing wrong with evaluating the model with proxy, in which case the model is to fit the proxy. But for molecule design, we should not ignore the gap between the proxy model and *oracle*, *i.e.*, the expensive computation. To this end, we propose two kinds of evaluation. Specifically, we first construct a dataset $\mathcal{D}_L$ with $300k$ molecules as in Bengio et al. (2021). A proxy model $\mathcal{P}_D$ is trained on $\mathcal{D}_L$ to serve as *oracle*, since we are unable to access the real *oracle* due to the expensive computation. We design two settings for evaluation: 1) **Fully trained:** In this case, we have $\mathcal{D}_L$ as an offline dataset and train our model on it. Online models are trained with $\mathcal{P}_D$. $\mathcal{P}_D$ is also employed to evaluate all molecule design models. 2) **Data scarcity:** In this case, a small dataset $\mathcal{D}_S$ containing about $14k$ molecules is generated by a well-trained generative model. COFlowNet is trained on the small dataset $\mathcal{D}_S$. Also, we have a weak proxy model $\mathcal{P}_W$ trained on $\mathcal{D}_S$, and online models are trained with $\mathcal{P}_W$. Similarly, $\mathcal{P}_D$ is employed to evaluate all molecule design models.

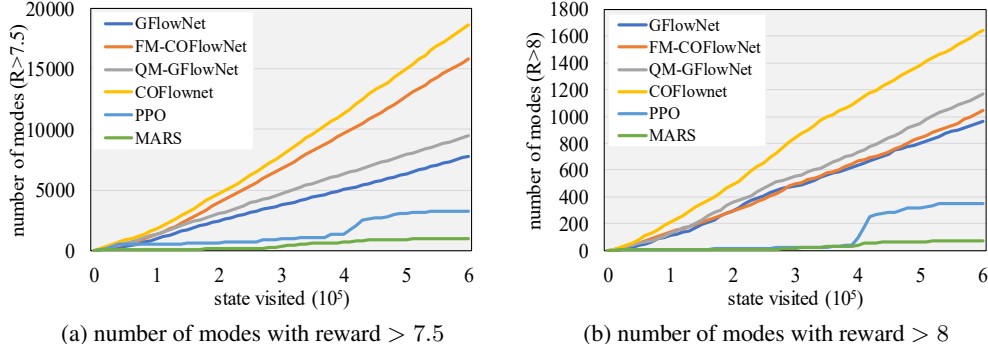

(a) number of modes with reward $> 7.5$          (b) number of modes with reward $> 8$

Figure 5: **Experimental result on Molecule Design with a large dataset.**

### 4.2.3 RESULT

We compare COFlowNet with two popular flow-based baselines, vanilla GFlowNet (Bengio et al., 2021) and QM-GFlowNet (Zhang et al., 2023d). Two more baselines, MARS (Xie et al., 2021) and PPO (Schulman et al., 2017), are involved here to compare COFlowNet with non-flow-based methods.

Noting that MARS and PPO require a fully-trained proxy model to provide accurate reward, we only report their performance under the setting of **Fully trained**. Additionally, we introduce *COFlowNet w/o* as an offline baseline, which removes the proposed conservative regularization term and is employed to substantiate our assertion that the regularization term enhances our model's performance. For the effectiveness comparison, we consider the average reward of the top $k = \{10, 100, 1000\}$ candidates. Such a metric indicates the ability of models to generate high-score candidates.

As reported in Table. 1, MARS performs significantly worse than flow-based models, including our COFlowNet. While PPO generates candidates with scores comparable to those of flow-based models, its performance in terms of diversity is markedly inferior, which will be shown in the following. The performance gap between COFlowNet and *COFlowNet w/o* validates the effectiveness of the proposed conservative regularization term on unsupported flows. QM-GFlowNet achieves the best performance in generating high-score candidates. Online models, QM-GFlowNet and GFlowNet, possess better exploration capabilities through interaction with proxy models, while Offline models can solely learn from collected data, where the state space is limited. Consequently, given a good proxy model, offline models can hardly outperform online models. The performance gap between GFlowNet and *COFlowNet w/o* also supports the analysis. Surprisingly, COFlowNet can beat vanilla GFlowNet on most of the metrics, especially when data is scarce to train a strong proxy model. Considering the above analysis between online and offline models, such results further validate the effectiveness of the proposed method. Meanwhile, as demonstrated in Table. 1, when the proxy model is suboptimal, with limited available data, the performance of QM-GFlowNet deteriorates more rapidly compared to COFlowNet.

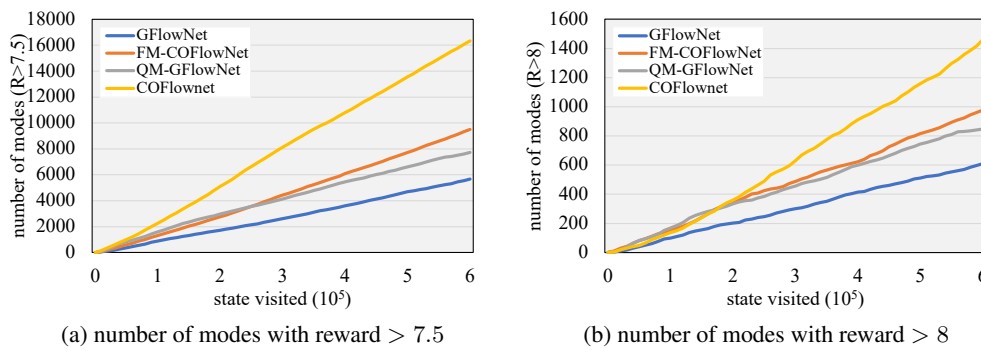

(a) number of modes with reward $> 7.5$      (b) number of modes with reward $> 8$

Figure 6: **Experimental result on Molecule Design with a small dataset.**

We further compare the diversity of candidates generated by different models. Considering that we introduce quantile matching loss for improving diversity, we here replace *COFlowNet w/o* with *FM-COFlowNet*, which utilizes flow matching loss rather than quantile matching loss. As shown in Fig. 5 and Fig. 6, COFlowNet achieves the best diversity. MARS and PPO show poor diversity performance. Notably, COFlowNet generates nearly 20 times as many candidate modes as MARS, demonstrating its superior ability to explore diverse candidates. The inferior diversity performance of MARS and PPO denotes that they are generating similar candidates and overfit the proxy model. The diversity gaps between FM-COFlowNet and COFlowNet, GFlowNet and QM-GFlowNet indicate the effectiveness of quantile matching loss in improving the diversity of candidates. Specifically, applying quantile matching loss leads to $10\%$ to $20\%$ improvement in diversity.

## 5   CONCLUSION

This paper extends GFlowNets to offline scenarios, especially for applications where evaluating a candidate is quite expensive and historical data has been collected. To take full utilization of the offline data, we define unsupported flows. By regularizing the unsupported flows, we prevent the model from making uncertain exploration of state space, thus generating candidates with higher scores. Additionally, to improve the character of GFlowNet in generating diverse candidates, we introduce a quantile matching algorithm. By modifying the regularization of unsupported flows into a quantile version, we finally propose the conservative offline GFlowNet, called COFlowNet.

## 6 ACKNOWLEDGEMENT

This paper is partially supported by the National Natural Science Foundation of China (No.12227901), the Natural Science Foundation of China Youth Project (No.62402472), and the Natural Science Foundation of Jiangsu Province of China Youth Project (No.BK20240461, No.BK20240460).

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

## A  TIME CONSUMPTION

In this part, we compare the training cost between the proposed method with online baselines. Compared to online models, COFlowNet is sampling-free during training, resulting in a reduction in training time. In Table. 2, we report the average training time per epoch for both COFlowNet and the baseline models. Since online models sample batches during training while offline models are trained on batched states, we report the time of online and offline models training on the same number of states ( *i.e.*, the number of states in the offline dataset), which we term as an epoch. The reported times reflect the average duration for 100 epochs. It is worth noting that we exclude the online evaluation time consumption of the online model (called oracle or proxy model) and the time consumption of training proxy models. As shown in Table. 2, the offline version of vanilla GFlowNet, *i.e.*, FM-COFlowNet, takes approximately $1/3$ less time to train than GFlowNet. Even with a more complex matching method, COFlowNet remains faster than GFlowNet and is approximately twice as fast as QM-GFlowNet. This demonstrates that COFlowNet offers significant efficiency advantages in terms of time cost.

| Model | GFlowNet | QM-GFlowNet | FM-COFlowNet | COFlowNet |
|---|---|---|---|---|
| Training time (s/epoch) | 1.594 | 2.904 | 1.055 | 1.521 |

Table 2: **Comparison of training time consumption.** Darker blue denotes the best result and lighter blue denotes the second best.

## B  EXPERIMENTS ON MORE TASKS

In Sec. 4, we have evaluated COFlowNet on two distinct tasks: the Hypergrid task and the Molecule Design task. Furthermore, our method is inherently generalizable to any domain where GFlowNets are applicable. To further validate the efficiency of COFlowNet beyond these tasks, we have extended our experiments to other tasks.

## B.1 ANTI-MICROBIAL PEPTIDE DESIGN

To further validate the efficiency of COFlowNet beyond these tasks, we have extended our experiments to include the **Anti-Microbial Peptide Design** task (Pirtskhalava et al., 2021). In this task, the objective is to generate peptides (short protein sequences) with anti-microbial properties, where actions involve selecting amino acids from a predefined set with 20 elements and the longest sequence is with 50 elements. For a given model, we have $D$ denoting the set of generated candidates with top 100 scores, and evaluate the methods using the following three metrics,

**Performance:** The average score/reward of the top 100 generated candidates.

$$\text{Performance}(D) = \frac{\sum_{x \in D} \text{reward}(x)}{|D|} \tag{16}$$

**Diversity:** The average pairwise distance among the top 100 candidates.

$$\text{Diversity}(D) = \frac{\sum_{x_i, x_j \in D} \text{d}(x_i, x_j)}{|D|(|D| - 1)} \tag{17}$$

where $\text{d}(\cdot, \cdot)$ is the Levenshtein distance between two sequences.
**Novelty:** The average distance between the top 100 candidates and known peptides, indicating the ability to generate new peptides.

$$\text{Novelty}(D) = \frac{\sum_{x_i \in D} \min_{s_j \in D_0} \text{d}(x_i, s_j)}{|D|} \tag{18}$$

where $D_0$ is the dataset used for training the proxy model.

| Model | Performance | Novelty | Diversity |
|---|---|---|---|
| QM-GFlowNet | 0.895 | 29.12 | 12.14 |
| GFlowNet | 0.868 | 15.72 | 11.32 |
| COFlowNet w/o | 0.788 | 25.68 | 10.43 |
| FM-COFlowNet | 0.853 | 28.53 | 13.44 |
| COFlowNet | 0.878 | 28.88 | 12.45 |

Table 3: **The average reward of the top $k$ candidates.** Darker blue denotes the best result and lighter blue denotes the second best.

The experimental results are summarized in Table. 3, showing that COFlowNet achieves superior performance across all three metrics, comparable to advanced online models despite being trained offline. These results further substantiate COFlowNet's capability to generate high-quality and diverse candidates across various tasks.
In summary, our COFlowNet demonstrates consistent and significant improvements over other methods across a diverse range of tasks, highlighting its broad applicability and efficiency.

## B.2 ITEM RECOMMENDATION

To further demonstrate the effectiveness of COFlowNet in diverse domains, we have extended our experiments to another classic and practical task in the domain of online businesses, **item recommendation**, inspired by the literature Liu et al. (2023b). In this task, for a given user $u$, models are required to sample trajectories, where each state corresponds to an item. Notably, instead of calculating rewards on the terminal states, we calculate rewards based on the whole trajectories in this task, which correspond to lists of items. This task aims to provide diverse recommendations with high quality, which is exactly a good application of the problem studied in this paper.

**Experimental setting.** The experiment is conducted on *ML1M* dataset, a subset of the MovieLens dataset[2]. We report three key metrics for evaluation: average reward (**Avg.R**), max reward (**Max.R**), and **Coverage**. We employ the online user environment proposed in Liu et al. (2023b) to calculate

---

[2]The dataset is available at `https://grouplens.org/datasets/movielens/1m/`

the preference of users to items, denoted as $s(u, a)$, which returns the user $u$'s response score of item $a$. Given a list of items $A$ and a user $u$, the **reward** of $A$ is calculated as

$$R(u, A) = \frac{1}{|A|} \sum_{a \in A} s(u, a).$$

In order to evaluate the *diversity* of candidates generated by the model, we employ the **Coverage** metric, which describes the number of distinct items exposed in a batch of generated lists of items.

**Experimental results and analysis.** The results are summarized in Table 4. According to the results, COFlowNet achieves similar performance to QM-GFlowNet in generating high-score candidates. Considering the inherent distinctions between online and offline models, the small performance gap between COFlowNet and QM-GFlowNet is predictable and acceptable. However, COFlowNet is able to generate candidates with more diversity than QM-GFlowNet, as evidenced by the Coverage metric. The Avg.R and Max.R gaps between *COFlowNet w/o* and COFlowNet further demonstrate the effectiveness of the proposed conservative constraint. The poor performance of *COFlowNet w/o* indicates that it is difficult to develop an effective offline model. Our proposed conservative constraint enables effective offline training. Additionally, COFlowNet, equipped with the quantile matching mechanism, outperforms FM-COFlowNet across all three metrics, validating the effectiveness of the proposed conservative quantile matching (CQM) objective introduced in Eq. 15.

| Model | Avg.R | Max.R | Coverage |
|---|---|---|---|
| GFlowNet | 1.996 | 2.832 | 47.95 |
| QM-GFlowNet | 2.016 | 2.865 | 65.61 |
| COFlowNet w/o | 1.280 | 2.692 | 115.1 |
| FM-COFlowNet | 1.664 | 2.800 | 62.15 |
| **COFlowNet** | **1.998** | **2.850** | **109.35** |

Table 4: Experimental results of item recommendation task.

Therefore, these results collectively demonstrate that **COFlowNet exhibits excellent offline learning capabilities and consistent effectiveness across various tasks,** and contributes to the research community in diverse valuable and practical domains.

