# OpenReview forum: "COFlowNet: Conservative Constraints on Flows Enable High-Quality Candidate Generation"
_ICLR.cc/2025/Conference — ICLR 2025 Poster_

### Official Review · Reviewer_7QzZ · 2024-10-25

**Soundness:** 3
**Presentation:** 3
**Contribution:** 3
**Rating:** 6
**Confidence:** 2

**Summary:**

In this paper, the author introduces COFlowNet, an offline Generative Flow Network (GFlowNet) designed to generate high-quality and diverse candidates without relying on online evaluations. The effectiveness of COFlowNet is validated through experiments on commonly used datasets.

**Strengths:**

+ Adapting GFlowNets for offline scenarios is a valuable idea.
+ The proposed method shows superiority over baseline models like QM-GFlowNet and FM-GFlowNet.

**Weaknesses:**

1. The authors conduct experiments on Hypergrid and molecule design tasks. Could the proposed method be generalized to more tasks and real applications? It is recommended to conduct experiments on additional tasks.

2. How much time and computational cost are saved by training in the offline setting? Could the authors compare the training cost between the proposed method and traditional methods?

3. It would be better to use \textbf{} for the first sentence in figure and table captions to indicate their targets clearly.

4. It would be  better to indicate the meaning of 'mixed', 'expert', and 'random' in Figure4.

5. Do number of modes and ℓ1 error are commonly used in evaluation of GFLowNets? Are there more metrics to measure the performance?

**Questions:**

See weakness

---

> ### Author Response · Authors · 2024-11-22
> **Response to Reviewer 7QzZ (1/2)**
>
> Dear Reviewer 7QzZ,
>
> Thank you very much for your valuable comments, which are crucial to the improvement of our paper. We would clarify your concerns point by point.
>
> ------
>
> >**Weaknesses 1:** The authors conduct experiments on Hypergrid and molecule design tasks. Could the proposed method be generalized to more tasks and real applications? It is recommended to conduct experiments on additional tasks.
>
> **Response:**
>
> We sincerely appreciate your insightful comments regarding the generalization of COFlowNet to more tasks. Based on your valuable suggestions, we have extended our experiments to include the **Anti-Microbial Peptide Design** [1] task. In this task, the goal is to generate peptides (short protein sequences) with anti-microbial properties, where actions involve selecting amino acids from a predefined set. We evaluate the methods using the following key metrics:
>
> - **Performance:** The average score/reward of the top 100 generated candidates.
> - **Diversity:** The average pairwise distance among the top 100 candidates.
> - **Novelty:** The average distance between the top 100 candidates and known peptides, indicating the ability to generate new peptides.
>
> The experimental results, summarized in the table below, show that COFlowNet achieves superior performance across all three metrics, comparable to advanced online models despite being trained offline. *These results further validate COFlowNet's ability to generate high-quality and diverse candidates across various tasks.*
>
> | Model         | Performance | Novelty   | Diversity |
> | ------------- | ----------- | --------- | --------- |
> | QM-GFLOWNET   | 0.895       | 29.12     | 12.14     |
> | GFlowNet      | 0.868       | 15.72     | 11.32     |
> | COFlowNet w/o | 0.788       | 25.68     | 10.43     |
> | FM-COFlowNet  | 0.853       | 28.53     | 13.44     |
> | **COFlowNet** | **0.878**   | **28.88** | **12.45** |
>
> We will incorporate these results and a detailed analysis into the revised manuscript to thoroughly address this concern. Additionally, *we are actively working on conducting further generalization experiments across a broader range of tasks to provide a more comprehensive evaluation of COFlowNet.* We will share updated experimental results with you as soon as they become available.
>
> [1] Pirtskhalava, Malak, et al. "DBAASP v3: database of antimicrobial/cytotoxic activity and structure of peptides as a resource for development of new therapeutics." Nucleic acids research 49.D1 (2021): D288-D297.
>
> ------
>
> >**Weaknesses 2:** How much time and computational cost are saved by training in the offline setting? Could the authors compare the training cost between the proposed method and traditional methods?
>
> **Response:**
>
> Thank you for your thoughtful question regarding the time cost savings achieved by training in the offline setting. *Compared to online models, COFlowNet is sampling-free during training, resulting in a reduction in training time.* In the table below, we report the average training time per epoch for both COFlowNet and the baseline models. Since online models sample batches during training while offline models are trained on batched states, we report the time of online and offline models training on the same number of states (i.e., the number of states in the offline dataset), which we term as an epoch. The reported times reflect the average duration for 100 epochs. It is worth noting that we exclude the online evaluation time consumption of the online model (calling oracle or proxy model) and the time consumption of training proxy models.
>
> | Model         | Training time (s/epoch)   |
> | --------      | --------                  |
> | GFlowNet      | 1.594                     |
> | QM-GFlowNet   | 2.904                     |
> | FM-COFlowNet  | 1.055                     |
> | **COFlowNet**    | **1.521**                     |
>
> As shown in the table, the offline version of vanilla GFlowNet, *i.e.*, FM-COFlowNet, takes approximately 1/3 less time to train than GFlowNet. Even with a more complex matching method, COFlowNet remains faster than GFlowNet and is approximately twice as fast as QM-GFlowNet. *This demonstrates that COFlowNet offers significant efficiency advantages in terms of time cost.*

---

> > ### Author Response · Authors · 2024-11-22
> > **Response to Reviewer 7QzZ (2/2)**
> >
> > >**Weaknesses 3:** It would be better to use \textbf{} for the first sentence in figure and table captions to indicate their targets clearly.
> >
> > **Response:**
> >
> > Thank you for your valuable suggestion on improving the readability of our manuscript. We will emphasize the subjects of figures and tables by using `\textbf{}` for the first sentence in figure and table captions to clearly highlight their targets in the revised manuscript.
> >
> > ------
> >
> > >**Weaknesses 4:** It would be better to indicate the meaning of 'mixed', 'expert', and 'random' in Figure4.
> >
> > **Response:**
> >
> > Thank you for your helpful suggestions, which are crucial for improving the presentation of our manuscript. The terms "mixed," "expert," and "random" refer to three strategies used to construct the offline dataset for training COFlowNet on the Hypergrid task (Section 4.1.2). Their specific meanings are as follows:
> >
> > - **Expert:** employ an online GFlowNet to generate an offline dataset with 2 × 10⁴ trajectories.
> > - **Random:** randomly generate an offline dataset with 2 × 10⁴ trajectories.
> > - **Mixed:** take 10⁴ trajectories from **Expert** and 10⁴ trajectories from **Random**.
> >
> > We have incorporated these definitions into Figure 4 as per your suggestion and will update the figure accordingly in the revised manuscript.
> >
> > ------
> >
> > >**Weaknesses 5:** Do number of modes and ℓ1 error are commonly used in evaluation of GFLowNets? Are there more metrics to measure the performance?
> >
> > **Response:**
> >
> > Thank you for raising this important point regarding the evaluation metrics. *The metrics used in our manuscript are commonly adopted for evaluating GFlowNets,* and their selection depends on the specific tasks. Below, we clarify the rationale behind the choice of metrics for the two tasks considered in our study: Hypergrid and Molecule Design.
> >
> > 1. **Hypergrid Task:**
> >    In this task, the complete state space has a finite size, allowing us to enumerate all states and calculate the ground-truth distribution. Our focus is on how well the model fits the ground-truth distribution (*i.e.*, whether the flow-based model generates candidates proportional to their scores). For this reason, we use the ℓ1 error as a metric for evaluation.
> > 2. **Molecule Design Task:**
> >    In molecule design, the primary interest lies in generating a diverse set of high-score molecules. We evaluate the models using the number of modes with rewards greater than thresholds (*e.g.*, 7.5 or 8). By setting this reward constraint, we focus only on high-score molecules.
> >
> > Overall, the evaluation of flow-based models typically focuses on two key aspects:
> >    - **Performance:** The model's ability to generate high-score candidates as expected.
> >    - **Diversity:** The model's capacity to generate a diverse set of candidates.
> >
> > The metrics used in our manuscript align well with these considerations and provide a comprehensive assessment of our model's contributions in generating both high-score and diverse candidates.
> >
> > We will include a detailed explanation of the metric selection in the revised manuscript. Thank you again for your valuable feedback and continued support of our work!

---

> > > ### Comment · Reviewer_7QzZ · 2024-11-24
> > >
> > > I appreciate the authors' detailed responses, which have addressed my concerns. I am open to improving the score if other reviewers are in agreement.

---

> > > > ### Author Response · Authors · 2024-11-24
> > > > **Thanks for your positive feedback!**
> > > >
> > > > Dear Reviewer 7QzZ,
> > > >
> > > > Thank you for thoroughly reviewing our responses in a timely manner. We are honored that our responses have successfully addressed your concerns. We have incorporated all of your constructive suggestions into the revised manuscript, significantly improving its clarity and overall quality. In addition, inspired by your insightful comments, we are making unremitting efforts to strengthen the persuasive and impactful nature of this work by conducting additional experiments, to improve the quality of the manuscript and its contribution to the community even further.
> > > >
> > > > We deeply appreciate your prompt response and are greatly encouraged by your positive feedback regarding your willingness to raise the score, which is a tremendous recognition for us. Thank you once again for your valuable efforts and continued support!
> > > >
> > > >
> > > > Best regards,
> > > >
> > > > Authors of submission 9900

---

> > > > ### Author Response · Authors · 2024-12-02
> > > > **Kind Follow-Up: Discussion Period Nearing Closure**
> > > >
> > > > Dear Reviewer 7QzZ,
> > > >
> > > > We hope this message finds you well! We appreciate your keen interest and high recognition of our work, including its **valuable idea** and **superior performance**.
> > > >
> > > > **We are honored that our detailed responses have addressed your concerns,** as reflected in your positive feedback. During the rebuttal period, we deeply value your constructive suggestions and **have made comprehensive revisions to the manuscript, significantly enhancing its overall quality**. Both the revised manuscript and updated code have been uploaded for your review.
> > > >
> > > > We are committed to **developing an offline model capable of generating more diverse candidates with high quality,** extending Generative Flow Networks to various application scenarios, and providing novel insights and effective solutions to advance the field. We truly treasure the opportunity to discuss with excellent peers like you, which not only makes our work well improved, but also opens up our research horizons to better contribute to the community.
> > > >
> > > > As the rebuttal period draws to a close, we welcome any further feedback you may have and would be delighted to learn from your insights. Meantime, **we sincerely hope that you will kindly consider improving your score,** as it is a recognition of the enhancements made to our work and a great encouragement to our ongoing efforts. Thank you once again for your valuable time and continued support. Wishing you all the best in your endeavors!
> > > >
> > > > Best regards,
> > > >
> > > > Authors of submission 9900

---

> ### Author Response · Authors · 2024-11-27
> **Latest efforts to share with you!**
>
> Dear Reviewer 7QzZ,
>
> We hope that everything goes well for you! Thank you for your interest and recognition of our work. Guided by your insightful comments, *we have incorporated all of your constructive suggestions into the revised manuscript and updated our code repository to greatly improve its clarity and overall quality.* Additionally, we have been making unremitting efforts to strengthen the persuasive and impactful nature of this work across diverse domains by conducting additional experiments. Here, *we are excited to share our latest experimental results with you,* and the details of the additional experiment are as follows:
>
> **New Task:** We have introduced another classic and practical task in the domain of online businesses, **item recommendation**, inspired by the literature [1]. In this task, for a given user $u$, models are required to sample trajectories, where each state corresponds to an item. Notably, *Instead of calculating rewards on the terminal states, we calculate rewards based on the whole trajectories in this task, which correspond to lists of items.* This task aims to provide diverse recommendations with high quality, which is exactly a good application of the problem studied in this paper.
>
> **Experimental setting:** The experiment is conducted on *ML1M* dataset, a subset of  MovieLens dataset (available at https://grouplens.org/datasets/movielens/1m/). We report three key metrics for evaluation: average reward (**Avg.R**), max reward (**Max.R**), and **Coverage**. We employ the online user environment proposed in [1] to calculate the preference of users to items, denoted as $s(u, a)$, which returns the user $u$'s response score of item $a$. Given a list of items $A$ and a user $u$, the **reward** of $A$ is calculated as $R(u, A) = \frac{1}{|A|}\sum_{a\in A} s(u, a)$. In order to evaluate the *diversity* of candidates generated by model, we employ **Coverage** metric which describes the number of distinct items exposed in a batch of generated lists of items. More details about this experiment have been updated in the revised version of our manuscript.
>
> **Experimental results & analysis:** The results are summarized in the table below. COFlowNet achieves similar performance to QM-GFlowNet in generating high-score candidates. Considering the inherent distinctions between online and offline models, the small performance gap between COFlowNet and QM-GFlowNet is predictable and acceptable. However, COFlowNet is able to generate candidates with more diversity than QM-GFlowNet, as evidenced by the Coverage metric. The Avg.R and Max.R gaps between *COFlowNet w/o* and COFlowNet further demonstrate the effectiveness of the proposed conservative constraint. And the poor performance of *COFlowNet w/o* indicates that it is difficult to develop an effective offline model. Our proposed conservative constraint enables effective offline training. Additionally, COFlowNet, equipped with the quantile matching mechanism, outperforms FM-COFlowNet across all three metrics, validating the effectiveness of the proposed conservative quantile matching (CQM) objective introduced in Eq. (15).
>
>
> | Model         | Avg.R     | Max.R     | Coverage   |
> | ------------- | --------- | --------- | ---------- |
> | GFlowNet      | 1.996     | 2.832     | 47.95      |
> | QM-GFlowNet   | 2.016     | 2.865     | 65.61      |
> | COFlowNet w/o | 1.280     | 2.692     | 115.1      |
> | FM-COFlowNet  | 1.664     | 2.800     | 62.15      |
> | **COFlowNet** | **1.998** | **2.850** | **109.35** |
>
> Therefore, these results collectively demonstrate that **COFlowNet exhibits excellent offline learning capabilities and consistent effectiveness across various tasks.** We have incorporated these results and detailed analyses into the revised manuscript to further enhance its quality and contribution to the community.
>
> We are honored that our responses have successfully addressed your concerns and deeply appreciate the opportunity to learn from you. At the same time, *we sincerely hope that you will kindly update the score due to the limited time available,* which would be a great encouragement to us. Thank you once again for your valuable time and continued support!
>
> Best regards,
>
> Authors of submission 9900
>
> ------
>
> [1] Liu, Shuchang, et al. "Generative flow network for listwise recommendation." *Proceedings of the 29th ACM SIGKDD Conference on Knowledge Discovery and Data Mining*. 2023.

---

> ### Author Response · Authors · 2024-12-02
> **A Gentle Reminder: Discussion Period Nearing Closure**
>
> Dear Reviewer 7QzZ,
>
> We sincerely appreciate your valued efforts and continued support of our work. We have carefully responded to each of your comments and thoroughly revised the manuscript accordingly. **We are deeply honored that our detailed responses have addressed your concerns during the rebuttal process.**
>
> As the discussion period is coming to a close on **December 2nd (today)**, and If you have no further comments, **we kindly remind you to consider improving your score.**
>
> Thank you again for your time and consideration!
>
> Best regards,
>
> Authors of submission 9900

---

### Official Review · Reviewer_9yMW · 2024-11-04

**Soundness:** 3
**Presentation:** 3
**Contribution:** 2
**Rating:** 5
**Confidence:** 4

**Summary:**

The paper present a method to the problem of offline GFlowNet training. Many conventional approaches require approximating training scores/rewards of candidate model outputs to provide a variety of generated candidate solutions, however, this approach is sensitive to out-of-distribution data. In this work the authors remove any reliance on proxy models for estimating the candidate to reward optimization objective for GFlowNet training. The authors propose to constrain the actions that a GFlowNet model may take to generate an output sample such that only actions supported by the training data are permissible. The authors show that their flow regularization only positively affects training on the supported flow paths, and that their approach is truly offline and can improve candidate model diversity.

**Strengths:**

The authors demonstrate originality in their work by clearly outlining their research problems in the context of other popular works and systematically laying out solutions for them with corresponding theorems and proofs. While maintaining an easy-to-read and high-level dialogue in addressing their approach’s significance, aim, and scope, the authors also manage to spare no mathematical details when necessary. Theorem 1 and the writing in sections 3.3 and 4 show formally the details required to both implement and understand the proposed approach.

**Weaknesses:**

In theorem 2, the use of the phrase “be more close to policy of offline dataset” seems too imprecise to be in the theorem statement. Since the proof is rather short this could perhaps be moved to the end of the proof, so that reader’s still get the high-level point that the author is trying to convey. Additionally, the conclusion of the proof for theorem 2 seems to be absent, although the technical/algebraic details seem sound.

Although the authors effectively demonstrate the efficacy of COFlowNet on the Molecule Design problem, providing experimental results on datasets from other domains would help show the success of their method beyond a shadow of a doubt.

**Questions:**

1. No references are provided for vanilla GFlowNet and QM-GFlowNet. There is no justification that these are the state-of-the-art methods.
2. Although COFlowNet w/o shows that the regularization term enhances performance, but can not beat the competing method QM-GFlowNet. The experiment results are weak. If the experiment results can not be improved, please provide a more in-depth analysis of where COFlowNet falls short compared to QM-GFlowNet.
3. Are there other potential ways of increasing diversity during GFlowNet regularization that were explored aside from the quantile matching algorithm?
4. Can COFlowNet be easily extended to other domains where RL is popular, like game playing and autonomous vehicle driving for example?

---

> ### Author Response · Authors · 2024-11-22
> **Response to Reviewer 9yMW (1/4)**
>
> Dear Reviewer 9yMW,
>
> Thank you very much for your valuable comments, which are crucial to the improvement of our paper. We would clarify your concerns point by point.
>
> ------
>
> >**Weaknesses 1:** In theorem 2, the use of the phrase “be more close to policy of offline dataset” seems too imprecise to be in the theorem statement. Since the proof is rather short this could perhaps be moved to the end of the proof, so that reader’s still get the high-level point that the author is trying to convey. Additionally, the conclusion of the proof for theorem 2 seems to be absent, although the technical/algebraic details seem sound.
>
> **Response:**
>
> Thank you very much for pointing out this shortcoming. Based on your constructive suggestions, we have carefully revised the formulation of Theorem 2 and its proof to improve clarity and rigor, and to ensure the point we are conveying is more precise. The revised version of Theorem 2 and its proof is as follows:
>
> **Theorem 2.** Given any state, minimizing its inflow from unsupported states will enlarge the inflow from supported states.
>
> **Proof.** For any state $s$, for example, $s_4$ in Figure 3, we have that the total inflows equal the total outflows:
> $$
> F(s_1, s_4) + F(s_2, s_4) + F(s_3, s_4) = F(s_4, S_c)
> $$
> where $S_c$ denotes all the children of $s_4$ and $F(s_4, S_c)$ is the sum of outflows from $s_4$ to its children. The total reward (flow) in the training data is determined, as all terminal states are given by the data, and the reward can only be obtained at terminal states. Therefore, the outflow $F(s_4, S_c)$ is constant and denoted as $F_o$. Furthermore, the inflow from supported states $F(s_1, s_4) + F(s_2, s_4)$ is equal to $F_o - F(s_3, s_4)$. Since $F_o$ is constant, minimizing the inflow from unsupported states $F(s_3, s_4)$ will lead to an increase in the inflow from supported states $F(s_1, s_4) + F(s_2, s_4)$. This reasoning process can be easily extended to general conditions (with more supported and unsupported states).
>
>
> We will update our manuscript with this revised version of Theorem 2 and its proof in accordance with your valuable suggestions.
>
> ------
>
> >**Weaknesses 2:** Although the authors effectively demonstrate the efficacy of COFlowNet on the Molecule Design problem, providing experimental results on datasets from other domains would help show the success of their method beyond a shadow of a doubt.
>
> **Response:**
>
> Thank you for your insightful comments regarding the scope of tasks used to evaluate our proposed COFlowNet. In our manuscript, we have evaluated COFlowNet on two distinct tasks: the Hypergrid task (Section 4.1) and the Molecule Design task (Section 4.2). Both tasks are standard benchmarks for GFlowNets and demonstrate the efficacy of COFlowNet in typical application scenarios. Moreover, our method is inherently generalizable to any domain where GFlowNets are applicable.
>
> To further validate the efficacy of COFlowNet beyond these tasks, we have extended our experiments to include the **Anti-Microbial Peptide Design** task [1] based on your valuable suggestion. In this task, the objective is to generate peptides (short protein sequences) with anti-microbial properties, where actions involve selecting amino acids from a predefined set. We evaluate the methods using three key metrics:
>
> - **Performance:** The average score/reward of the top 100 generated candidates.
> - **Diversity:** The average pairwise distance among the top 100 candidates.
> - **Novelty:** The average distance between the top 100 candidates and known peptides, indicating the ability to generate new peptides.
>
> The experimental results, summarized in the table below, show that COFlowNet achieves superior performance across all three metrics, comparable to advanced online models despite being trained offline. These results further substantiate COFlowNet’s capability to generate high-quality and diverse candidates across various tasks.
>
> | Model         | Performance | Novelty   | Diversity |
> | ------------- | ----------- | --------- | --------- |
> | QM-GFLOWNET   | 0.895       | 29.12     | 12.14     |
> | GFlowNet      | 0.868       | 15.72     | 11.32     |
> | COFlowNet w/o | 0.788       | 25.68     | 10.43     |
> | FM-COFlowNet  | 0.853       | 28.53     | 13.44     |
> | **COFlowNet** | **0.878**   | **28.88** | **12.45** |
>
> In summary, *our COFlowNet demonstrates consistent and significant improvements over other methods across a diverse range of tasks, highlighting its broad applicability and effectiveness.* We will incorporate these results and a detailed analysis into the revised manuscript to thoroughly address this concern.
>
> Additionally, *we are actively working on conducting further generalization experiments across a broader range of tasks to provide a more comprehensive evaluation of COFlowNet.* We will share updated experimental results with you as soon as they become available.

---

> ### Author Response · Authors · 2024-11-22
> **Response to Reviewer 9yMW (2/4)**
>
> >**Questions 1:** No references are provided for vanilla GFlowNet and QM-GFlowNet. There is no justification that these are the state-of-the-art methods.
>
> **Response:**
>
> Thank you for pointing out this issue. We sincerely apologize for the oversight in not properly denoting the references for vanilla GFlowNet and QM-GFlowNet in the experimental section. While the references for vanilla GFlowNet [2] and QM-GFlowNet [3] are mentioned in other parts of our manuscript, we have now added them to the experimental section to enhance readability and rigor, as per your suggestion. Below are brief descriptions of these two models:
> - Vanilla GFlowNet [2] is based on a view of the generative process as a flow network, making it possible to handle the tricky case where different trajectories can yield the same final state, *e.g.*, there are many ways to sequentially add atoms to generate some molecular graph.
> - QM-GFlowNet [3] adopts a distributional paradigm for GFlowNets, turning each flow function into a distribution, thus providing more informative learning signals during training.
>
> Furthermore, QM-GFlowNet was made publicly available on arXiv in 2023 and is one of the most recent works in flow-based models. In the experimental section of the original paper, *the authors confirmed through extensive experiments that QM-GFlowNet is the online model that currently achieves state-of-the-art performance in various tasks, including the Hypergrid and molecule optimization tasks studied in our paper.* Therefore, we included these models in our experiment as advanced baselines for comparative analysis to evaluate our COFlowNet.
>
> We will add the appropriate citations to the revised manuscript and elaborate on the justification for selecting these models as state-of-the-art baselines. Thank you again for your valuable guidance, which has significantly contributed to improving the clarity and quality of our manuscript.
>
>
> ------
>
> >**Questions 2:** Although COFlowNet w/o shows that the regularization term enhances performance, but can not beat the competing method QM-GFlowNet. The experiment results are weak. If the experiment results can not be improved, please provide a more in-depth analysis of where COFlowNet falls short compared to QM-GFlowNet.
>
> **Response:**
>
> Thank you for your insightful comments. *The performance difference between COFlowNet and QM-GFlowNet reflects the inherent distinctions between online and offline models.* As such, it is challenging for an offline model like COFlowNet to surpass a state-of-the-art online model such as QM-GFlowNet. We provide an in-depth analysis of the reasons for this phenomenon: Online models possess better exploration capabilities through interaction with proxy models, while Offline models can solely learn from collected data, where the state space is limited. Consequently, given a good proxy model, offline models can hardly outperform online models. Nevertheless, as demonstrated in Table 1, when the proxy model is suboptimal (the amount of available data is limited), the performance of QM-GFlowNet deteriorates more rapidly compared to COFlowNet.
>
> Based on your constructive suggestions, we will include a more in-depth analysis of the experimental results in the revised manuscript to clarify the distinctions between online and offline models, as well as the reasons for the observed performance differences.

---

> > ### Author Response · Authors · 2024-11-22
> > **Response to Reviewer 9yMW (3/4)**
> >
> > >**Questions 3:** Are there other potential ways of increasing diversity during GFlowNet regularization that were explored aside from the quantile matching algorithm?
> >
> > **Response:**
> >
> > Thank you for your constructive comments and interest in our work. We indeed explored various potential approaches to increasing diversity during GFlowNet regularization, which we elaborate on below.
> >
> > GFlowNet training methods can generally be categorized into four types [4]: Flow Matching (FM), Detailed Balance(DB), Trajectory Balance (TB), and Quantile Matching (QM). Among these, TB is classified as a Monte Carlo method, while the others are temporal-difference (TD)-based methods. Theoretically, the COFlowNet constraint can be applied to any TD-based formulation of the GFlowNet objective. DB is primarily designed to reduce computation time and accelerate training convergence, and offers little benefit in increasing diversity. While the TB-based approaches are significantly faster, roughly five times quicker than the others, the TB loss optimizes the overall trajectory as a whole but may incur a large variance in the squared error term according to the literature [5]. In fact,  in molecule optimization tasks, the number of molecules found by the TB-based methods is even fewer than those discovered by traditional models like PPO and MARS, as demonstrated in QM-GFlowNet [3]. Furthermore, in other molecule generation tasks QM9 and sEH discussed in the literature [6], TB-GFlowNets consistently underperform relative to FM-GFlowNets in terms of overall effectiveness and diversity. Through a thorough investigation, we have identified Quantile Matching (QM) as the state-of-the-art method and most effective strategy for enhancing diversity, which achieves over 10 times the number of modes than TB method and nearly three times as many modes as the FM method in molecule design, as demonstrated in QM-GFlowNet [3]. Besides, QM provides more a informative learning signal for better generalization, reduces overestimation problems, and alleviates pseudo-uncertainty[3]. Considering these advantages, we integrated QM into our method to effectively enhance diversity.
> >
> > We will add a discussion on potential ways to increase diversity in the GFlowNet regularization in the revised version to help readers understand our work more clearly.

---

> > > ### Author Response · Authors · 2024-11-22
> > > **Response to Reviewer 9yMW (4/4)**
> > >
> > > >**Questions 4:** Can COFlowNet be easily extended to other domains where RL is popular, like game playing and autonomous vehicle driving for example?
> > >
> > > **Response:**
> > >
> > > Thank you for your thoughtful question. As mentioned in our **Response** to **Weakness 2**, our method is inherently generalizable to any domain where GFlowNets are applicable. In response, we have conducted additional experiments on the task of **Anti-Microbial Peptide Design**, which further validate the generalization and effectiveness of COFlowNet. The detailed experimental results and analysis are provided in our **Response** to **Weakness 2**.
> > >
> > > Regarding the domains you mentioned, such as game playing and autonomous vehicle driving, we conducted an in-depth investigation to assess their compatibility with GFlowNet-based models. For autonomous vehicle driving: GFlowNet-based models, including COFlowNet, have natural limitations that make them less suitable for autonomous vehicle driving. Specifically: i) The action space for GFlowNets must be discrete and not very large. Autonomous vehicle driving typically involves continuous control variables, such as speed and direction, which are not well-suited to the discrete action framework of GFlowNets. ii) The key property of GFlowNet-based models is their ability to generate candidates in proportion to their rewards. In autonomous driving, the goal is typically to determine a single optimal strategy rather than a series of possible operations, making GFlowNets less applicable to this domain. For game playing: It is feasible to extend COFlowNet to certain types of games where the action space is discrete, such as chess-like games. Indeed, there are existing works that explore the application of GFlowNet-based models to game playing. However, due to the limited time available during the discussion period, we may not be able to extend COFlowNet to the game tasks described in time, while we are committed to continuing to explore this direction. In addition, *we are also actively working on conducting further generalization experiments in other domains.* We will share updated experimental results as soon as they are available.
> > >
> > >
> > > We will update the experimental results and analyses of COFlowNet on other domains and provide a detailed discussion on the model's scope of application in the revised version. Thank you again for your constructive insights and continued support!
> > >
> > > ------
> > >
> > > [1] Pirtskhalava, Malak, et al. "DBAASP v3: database of antimicrobial/cytotoxic activity and structure of peptides as a resource for development of new therapeutics." Nucleic acids research 49.D1 (2021): D288-D297.
> > >
> > > [2] Bengio, Emmanuel, et al. "Flow network based generative models for non-iterative diverse candidate generation." Advances in Neural Information Processing Systems 34 (2021): 27381-27394.
> > >
> > > [3] Zhang, Dinghuai, et al. "Distributional gflownets with quantile flows." arXiv preprint arXiv:2302.05793 (2023).
> > >
> > > [4] Malkin, Nikolay, et al. "Trajectory balance: Improved credit assignment in gflownets." Advances in Neural Information Processing Systems 35 (2022): 5955-5967.
> > >
> > > [5] Madan, Kanika, et al. "Learning gflownets from partial episodes for improved convergence and stability." International Conference on Machine Learning. PMLR, 2023.
> > >
> > > [6] He, Haoran, et al. "Rectifying Reinforcement Learning for Reward Matching." arXiv preprint arXiv:2406.02213 (2024).

---

> > > > ### Comment · Reviewer_9yMW · 2024-11-26
> > > >
> > > > Thanks for addressing the comments with so much detail. The paper will have a better quality after these improvements.

---

> ### Author Response · Authors · 2024-11-27
> **Thanks for your positive feedback!**
>
> Dear Reviewer 9yMW,
>
> We are honored that our responses can address your comments. We have incorporated all of your constructive suggestions into the revised manuscript, significantly improving its clarity and overall quality. According to your constructive suggestions, we have been working on evaluating COFlowNet on more tasks to further demonstrate its effectiveness. Here, *we are excited to share our latest experimental results with you.* Details of the additional experiment are as follows:
>
> **New Task:** We have introduced another classic and practical task in the domain of online businesses, **item recommendation**, inspired by the literature [1]. In this task, for a given user $u$, models are required to sample trajectories, where each state corresponds to an item. Notably, *Instead of calculating rewards on the terminal states, we calculate rewards based on the whole trajectories in this task, which correspond to lists of items.* This task aims to provide diverse recommendations with high quality, which is exactly a good application of the problem studied in this paper.
>
> **Experimental setting:** The experiment is conducted on *ML1M* dataset, a subset of  MovieLens dataset (available at https://grouplens.org/datasets/movielens/1m/). We report three key metrics for evaluation: average reward (**Avg.R**), max reward (**Max.R**), and **Coverage**. We employ the online user environment proposed in [1] to calculate the preference of users to items, denoted as $s(u, a)$, which returns the user $u$'s response score of item $a$. Given a list of items $A$ and a user $u$, the **reward** of $A$ is calculated as $R(u, A) = \frac{1}{|A|}\sum_{a\in A} s(u, a)$. In order to evaluate the *diversity* of candidates generated by model, we employ **Coverage** metric which describes the number of distinct items exposed in a batch of generated lists of items. More details about this experiment have been updated in the revised version of our manuscript.
>
> **Experimental results & analysis:** The results are summarized in the table below. COFlowNet achieves similar performance to QM-GFlowNet in generating high-score candidates. Considering the inherent distinctions between online and offline models, the small performance gap between COFlowNet and QM-GFlowNet is predictable and acceptable. However, COFlowNet is able to generate candidates with more diversity than QM-GFlowNet, as evidenced by the Coverage metric. The Avg.R and Max.R gaps between *COFlowNet w/o* and COFlowNet further demonstrate the effectiveness of the proposed conservative constraint. And the poor performance of *COFlowNet w/o* indicates that it is difficult to develop an effective offline model. Our proposed conservative constraint enables effective offline training. Additionally, COFlowNet, equipped with the quantile matching mechanism, outperforms FM-COFlowNet across all three metrics, validating the effectiveness of the proposed conservative quantile matching (CQM) objective introduced in Eq. (15).
>
>
> | Model         | Avg.R     | Max.R     | Coverage   |
> | -----         | -----     | ------    | --------   |
> | GFlowNet      | 1.996     | 2.832     | 47.95      |
> | QM-GFlowNet   | 2.016     | 2.865     | 65.61      |
> | COFlowNet w/o | 1.280     | 2.692     | 115.1      |
> | FM-COFlowNet  | 1.664     | 2.800     | 62.15      |
> | **COFlowNet** | **1.998** | **2.850** | **109.35** |
>
> Therefore, these results collectively demonstrate that **COFlowNet exhibits excellent offline learning capabilities and consistent effectiveness across various tasks.** We have incorporated these results and detailed analyses into the revised manuscript to further enhance its quality and contribution to the community.
>
> If there is anything else you would like to discuss with us please feel free to reach out, we deeply appreciate the opportunity to learn from you. At the same time, *we sincerely hope that you will kindly reconsider your evaluation of our work due to the limited time available,* which would be a great encouragement to us. Thank you once again for your valuable efforts and continued support!
>
> Best regards,
>
> Authors of submission 9900
>
> ------
>
> [1] Liu, Shuchang, et al. "Generative flow network for listwise recommendation." *Proceedings of the 29th ACM SIGKDD Conference on Knowledge Discovery and Data Mining*. 2023.

---

> ### Author Response · Authors · 2024-12-01
> **Kind Follow-Up: Discussion Period Nearing Closure**
>
> Dear Reviewer 9yMW,
>
> We hope this message finds you well! We appreciate your keen interest and high recognition of our work, including its **originality**, **easy-to-read and high-level dialogue**, and **necessary mathematical details**.
>
> **We are honored that our responses have addressed your comments in great detail.** During the rebuttal period, we deeply value
>  each of your constructive suggestions and **have made comprehensive revisions to our manuscript, significantly enhancing its overall quality** as you can see. Both the revised manuscript and updated code have been uploaded for your review.
>
> We are committed to **developing an offline model capable of generating more diverse candidates with high quality,** extending Generative Flow Networks to various application scenarios, and providing novel insights and effective solutions to advance the field. We truly treasure the opportunity to discuss with excellent peers like you, which not only makes our work well improved, but also opens up our research horizons to better contribute to the community.
>
> As the rebuttal period draws to a close, we welcome any further feedback you may have and would be delighted to learn from your insights. Meanwhile, **we sincerely hope that you will kindly consider updating your score,** as it is a recognition of the enhancements made to our work and a great encouragement to our ongoing efforts. Thank you once again for your valuable time and continued support. Wishing you all the best in your endeavors!
>
> Best regards,
>
> Authors of submission 9900

---

> ### Author Response · Authors · 2024-12-02
> **A Gentle Reminder: Discussion Period Nearing Closure**
>
> Dear Reviewer 9yMW,
>
> We sincerely appreciate your valued efforts and continued support of our work.  **We have carefully responded to each of your comments and thoroughly revised the manuscript accordingly.**  We are deeply honored that our detailed responses have addressed your comments during the rebuttal process.
>
> As the discussion period is coming to a close on December 2nd, and If you have no further comments, **we kindly remind you to consider raising your score.**
>
> Thank you again for your time and consideration!
>
> Best regards,
>
> Authors of submission 9900

---

### Official Review · Reviewer_z4PM · 2024-11-04

**Soundness:** 4
**Presentation:** 3
**Contribution:** 2
**Rating:** 6
**Confidence:** 3

**Summary:**

This paper tackles the generation task, which aims to search for state transitions with high rewards. The fundamental task strongly impacts generating or searching for plausible candidates of sequences, where its possible applications include molecule design or path generation.
To this end, the authors have proposed COFlowNet, which focuses on finding high-reward state transitions by simply regularizing unsupported flows. The efficacy of the method is confirmed in Hypergrid and Molecule Design tasks.

**Strengths:**

**Strength 1:** The main strength of this work is the broad impact on any candidate-generation tasks.
- I feel that the method can be applied to any candidate-generation task, such as molecule design. Also, as pointed out in the paper, flow generation can be further used to search plausible flows to find the optimal decision-making in the RL study.

**Weaknesses:**

**Weakness 1:** The comparison with prior works and the evaluations seem limited.
- By referring the baseline paper of GFlowNet, it is compared with other non-flow-based approaches such as MCMC (MARS) and PPO. However, COFlowNet is only compared with GFlowNet, which makes it hard to figure out the quantitative gains of COFlowNet over MCMC and PPO. It is better to add these two baselines in the experiments.
- Moreover, only the main experiment considered in this paper, which includes the comparisons with others, is Molecule Design. (I have not published works for this topic) In literature, is there any other task to demonstrate the efficiency of the proposed method? A single demonstration seems to be limited to say the consistent and meaningful gains of COFlowNet over others.

**Minor Comment 1:** For consistency, please choose the one among COFlowNet or COFlownet.

**Questions:**

**Questions 1:** Regarding the weakness in literature, is there any other task to demonstrate the efficiency of the proposed method? If so, it would be much better to add one or two more results of COFlowNet.

---

> ### Author Response · Authors · 2024-11-22
> **Response to Reviewer z4PM (1/2)**
>
> Dear Reviewer z4PM,
>
> Thank you very much for your valuable comments, which are crucial to the improvement of our paper. We would clarify your concerns point by point.
>
> ------
>
> > **Weakness 1.1:** By referring the baseline paper of GFlowNet, it is compared with other non-flow-based approaches such as MCMC (MARS) and PPO. It is better to add these two baselines in the experiments.
>
> **Response:**
>
> We sincerely appreciate your constructive suggestion to include additional comparisons with MCMC (MARS) and PPO. Following your advice, we conducted new experiments to evaluate these baselines on the molecule design task. The results are summarized in the table below, providing a quantitative assessment of COFlowNet's performance relative to MCMC and PPO.
>
> The results reveal that MARS performs significantly worse than flow-based models, including our COFlowNet, in both the quality and diversity of the generated candidates. Notably, COFlowNet generates nearly 20 times as many candidate modes as MARS, demonstrating its superior ability to explore diverse candidates. While PPO generates candidates with scores comparable to those of flow-based models, its performance in terms of diversity is markedly inferior.
>
> These findings highlight the significant advantages of our COFlowNet over non-flow-based approaches like MCMC and PPO, particularly in diversity. We will include these results and a more detailed discussion in the revised manuscript to further substantiate the effectiveness of COFlowNet.
>
> | Model         | avg top 10 | avg top 100 | avg top 1000 | #modes (R>7.5) | #modes (R>8) |
> | ------------- | ---------- | ----------- | ------------ | -------------- | ------------ |
> | **MARS**      | 8.0778     | 7.8330      | 7.5992       | 1011           | 60           |
> | **PPO**       | 8.4249     | 8.3387      | 8.2555       | 3240           | 349          |
> | GFlowNet      | 8.5283     | 8.3539      | 8.1440       | 7780           | 967          |
> | QM-GFlowNet   | 8.5552     | 8.4019      | 8.1886       | 9480           | 1169         |
> | COFlowNet w/o | 8.4859     | 8.3278      | 8.1083       | 15815          | 1047         |
> | **COFlowNet** | 8.5029     | 8.3730      | 8.1693       | **18687**      | **1643**     |
>
> *Tips:* ''#modes" denotes the number of modes.
>
> ------
>
> > **Weakness 1.2:** In literature, is there any other task to demonstrate the efficiency of the proposed method?
>
> **Response:**
>
> Thank you for your insightful comments regarding the scope of tasks used to evaluate our proposed COFlowNet. In our manuscript, we have evaluated COFlowNet on two distinct tasks: the Hypergrid task (Section 4.1) and the Molecule Design task (Section 4.2). Both tasks are standard benchmarks for GFlowNets and demonstrate the effectiveness of COFlowNet in typical application scenarios. Furthermore, our method is inherently generalizable to any domain where GFlowNets are applicable.
>
> To further validate the efficiency of COFlowNet beyond these tasks, we have extended our experiments to include the **Anti-Microbial Peptide Design** task [1] based on your valuable suggestion. In this task, the objective is to generate peptides (short protein sequences) with anti-microbial properties, where actions involve selecting amino acids from a predefined set. We evaluate the methods using three key metrics:
>
> - **Performance:** The average score/reward of the top 100 generated candidates.
> - **Diversity:** The average pairwise distance among the top 100 candidates.
> - **Novelty:** The average distance between the top 100 candidates and known peptides, indicating the ability to generate new peptides.
>
> The experimental results, summarized in the table below, show that COFlowNet achieves superior performance across all three metrics, comparable to advanced online models despite being trained offline. These results further substantiate COFlowNet’s capability to generate high-quality and diverse candidates across various tasks.
>
> | Model         | Performance | Novelty   | Diversity |
> | ------------- | ----------- | --------- | --------- |
> | QM-GFLOWNET   | 0.895       | 29.12     | 12.14     |
> | GFlowNet      | 0.868       | 15.72     | 11.32     |
> | COFlowNet w/o | 0.788       | 25.68     | 10.43     |
> | FM-COFlowNet  | 0.853       | 28.53     | 13.44     |
> | **COFlowNet** | **0.878**   | **28.88** | **12.45** |
>
> In summary, *our COFlowNet demonstrates consistent and significant improvements over other methods across a diverse range of tasks, highlighting its broad applicability and efficiency.* We will incorporate these results and a detailed analysis into the revised manuscript to thoroughly address this concern.
>
> [1] Pirtskhalava, Malak, et al. "DBAASP v3: database of antimicrobial/cytotoxic activity and structure of peptides as a resource for development of new therapeutics." Nucleic acids research 49.D1 (2021): D288-D297.

---

> ### Author Response · Authors · 2024-11-22
> **Response to Reviewer z4PM (2/2)**
>
> >**Minor Comment 1:** Minor Comment: For consistency, please choose the one among COFlowNet or COFlownet.
>
> **Response:**
>
> Thank you for bringing this to our attention. The correct name of our model is **COFlowNet**, and we have thoroughly reviewed the entire manuscript to ensure consistency in its usage throughout the revised version.
>
> ------
>
> >**Questions 1:** Regarding the weakness in literature, is there any other task to demonstrate the efficiency of the proposed method? If so, it would be much better to add one or two more results of COFlowNet.
>
> **Response:**
>
> Thank you for highlighting this point. In response, we have conducted additional experiments on the task of **Anti-Microbial Peptide Design**, which further validate the efficiency of COFlowNet. The detailed experimental results and analysis are provided in our **Response** to **Weakness 1.1**.
>
> Additionally, *we are actively working on conducting validation experiments across a broader range of tasks to provide a more comprehensive evaluation of COFlowNet.* We will share updated experimental results with you as soon as they become available. We sincerely appreciate your continued interest and support in our work!

---

> ### Comment · Reviewer_z4PM · 2024-11-25
>
> - Thank you for the response with the additional evaluations. I agree that the proposed method, i.e., COFlowNet, and the related group of methods with graph flows outperform MARS and PPO. Also, COFlowNet has the strength of generating diverse candidates compared to other flow-based methods.
> - However, it is quite doubtful whether COFlowNet truly generates better candidates with higher scores than other flow-based methods. In the performance table provided during the rebuttal, COFlowNet is slightly below QM-GFlowNet and GFlowNet in 'avg top 10'.
> - Also, in the additional task, i.e., the Anti-Microbial Peptide Design task, COFlowNet's performance and novelty are inferior to those of QM-GFlowNet.
> - I acknowledge that COFlowNet is stronger in generating various candidates than others, but its novelty looks marginal from the perspectives of the quality of generated candidates and the algorithmic novelty beyond GFlowNet (COFlowNet is a regularized version of GFlowNet).
> - By considering the high impact of the work in many practical applications, I will not change my score.

---

> > ### Author Response · Authors · 2024-12-02
> > **A Gentle Reminder: Discussion Period Nearing Closure**
> >
> > Dear Reviewer z4PM,
> >
> > We deeply appreciate your valued efforts and continued support of our work.  **We have carefully responded to each of your comments and thoroughly revised the manuscript accordingly.**  We sincerely hope that your concerns have been addressed during the rebuttal process.
> >
> > As the discussion period is coming to a close on December 2nd, and If you have no further comments, **we kindly remind you to consider updating your score.**
> >
> > Thank you again for your time and consideration!
> >
> > Best regards,
> >
> > Authors of submission 9900

---

> ### Author Response · Authors · 2024-11-25
> **Thanks for your prompt feedback!**
>
> Dear Reviewer z4PM,
>
> Thank you for your further comments and for recognizing the strengths of our work, such as *the high impact in many practical applications*. We greatly appreciate your detailed feedback, and we would like to further clarify your concerns below.
>
> **COFlowNet is an offline flow-based model,** and can solely learn from collected data, which covers limited state space. However, online models, including GFlowNet and QM-GFlowNet, have a better ability to explore the whole state space by interacting with the proxy model. Consequently, in theory, given a good proxy model, offline models can hardly outperform online models in generating high-score candidates.
>
> **The performance difference between COFlowNet and QM-GFlowNet precisely reflects these inherent distinctions between online and offline models.** Such a phenomenon can also be observed when comparing GFlowNet and *COFlowNet w/o*, which corresponds to training GFlowNet in an offline paradigm. As such, it is challenging for an offline model like COFlowNet to surpass a state-of-the-art online model such as QM-GFlowNet. Nevertheless, as demonstrated in Table 1 in our manuscript, when the proxy model is suboptimal (the amount of available data is limited), the performance of QM-GFlowNet deteriorates more rapidly compared to COFlowNet.
>
> We would like to further clarify the key technical **contributions** of our work, which are in two folds, 1) we propose a **Conservative Constraint to enable effective offline training of GFlowNet**, and the effectiveness of this constraint can be validated by comparing *COFlowNet w/o* with COFlowNet. 2) we introduce a **quantile matching algorithm to enhance the diversity of generated candidates**.
>
> Due to the inherent differences between offline and online scenarios, we have to admit that the ability of COFlowNet to generate high-score candidates is slightly inferior to that of QM-GFlowNet, but our goal is not to outperform the online models on this point, which is also unpractical. *We have included the analysis of the experimental results in the revised manuscript to clarify the distinctions between online and offline models, as well as the reasons for the observed performance differences.* (Please refer to **Section 4.2.3** in the revised manuscript.)
>
> We sincerely hope that our response could address your concerns, and that you can kindly reconsider the contributions of our work. Thank you once again for your valuable efforts and continued support!
>
> Best regards,
>
> Authors of submission 9900

---

> ### Author Response · Authors · 2024-11-27
> **Sharing further experiments and discussions with you!**
>
> Dear Reviewer z4PM,
>
> According to your constructive suggestions, we have been working on evaluating COFlowNet on more tasks to further demonstrate its effectiveness. Here, *we are excited to share our latest experimental results with you.* Details of the additional experiment are as follows:
>
> **New Task:** We have introduced another classic and practical task in the domain of online businesses, **item recommendation**, inspired by the literature [1]. In this task, for a given user $u$, models are required to sample trajectories, where each state corresponds to an item. Notably, *Instead of calculating rewards on the terminal states, we calculate rewards based on the whole trajectories in this task, which correspond to lists of items.* This task aims to provide diverse recommendations with high quality, which is exactly a good application of the problem studied in this paper.
>
> **Experimental setting:** The experiment is conducted on *ML1M* dataset, a subset of  MovieLens dataset (available at https://grouplens.org/datasets/movielens/1m/). We report three key metrics for evaluation: average reward (**Avg.R**), max reward (**Max.R**), and **Coverage**. We employ the online user environment proposed in [1] to calculate the preference of users to items, denoted as $s(u, a)$, which returns the user $u$'s response score of item $a$. Given a list of items $A$ and a user $u$, the **reward** of $A$ is calculated as $R(u, A) = \frac{1}{|A|}\sum_{a\in A} s(u, a)$. In order to evaluate the *diversity* of candidates generated by model, we employ **Coverage** metric which describes the number of distinct items exposed in a batch of generated lists of items. More details about this experiment have been updated in the revised version of our manuscript.
>
> **Experimental results & analysis:** The results are summarized in the table below. COFlowNet achieves similar performance to QM-GFlowNet in generating high-score candidates. Considering the inherent distinctions between online and offline models, the small performance gap between COFlowNet and QM-GFlowNet is predictable and acceptable. However, COFlowNet is able to generate candidates with more diversity than QM-GFlowNet, as evidenced by the Coverage metric. The Avg.R and Max.R gaps between *COFlowNet w/o* and COFlowNet further demonstrate the effectiveness of the proposed conservative constraint. And the poor performance of *COFlowNet w/o* indicates that it is difficult to develop an effective offline model. Our proposed conservative constraint enables effective offline training. Additionally, COFlowNet, equipped with the quantile matching mechanism, outperforms FM-COFlowNet across all three metrics, validating the effectiveness of the proposed conservative quantile matching (CQM) objective introduced in Eq. (15).
>
>
> | Model         | Avg.R     | Max.R     | Coverage   |
> | -----         | -----     | ------    | --------   |
> | GFlowNet      | 1.996     | 2.832     | 47.95      |
> | QM-GFlowNet   | 2.016     | 2.865     | 65.61      |
> | COFlowNet w/o | 1.280     | 2.692     | 115.1      |
> | FM-COFlowNet  | 1.664     | 2.800     | 62.15      |
> | **COFlowNet** | **1.998** | **2.850** | **109.35** |
>
> Therefore, these results collectively demonstrate that **COFlowNet exhibits excellent offline learning capabilities and consistent effectiveness across various tasks.** We have incorporated these results and detailed analyses into the revised manuscript to further enhance its quality and contribution to the community.
>
> If there is anything else you would like to discuss with us please feel free to reach out, we deeply appreciate the opportunity to learn from you. At the same time, *we sincerely hope that you will kindly reconsider your evaluation of our work due to the limited time available,* which would be a great encouragement to us. Thank you once again for your valuable efforts and continued support!
>
> Best regards,
>
> Authors of submission 9900
>
> ------
>
> [1] Liu, Shuchang, et al. "Generative flow network for listwise recommendation." *Proceedings of the 29th ACM SIGKDD Conference on Knowledge Discovery and Data Mining*. 2023.

---

> ### Author Response · Authors · 2024-12-01
> **Kind Follow-Up: Discussion Period Nearing Closure**
>
> Dear Reviewer z4PM,
>
> We hope this message finds you well! Thank you very much for your keen interest and high recognition of our work. As you mentioned in your comments, one of the key strengths of our work is "**the high and broad impact in many practical applications.**" During the rebuttal period, we further validated the effectiveness of our proposed COFlowNet across a wider range of decision-making tasks, reaffirming your insightful point that "flow generation can be further used to search plausible flows to find the optimal decision-making in the RL study." We are committed to **developing an offline model capable of generating more diverse candidates with high quality,** extending Generative Flow Networks to various application scenarios, and providing novel insights and effective solutions to advance the field.
>
> We deeply value your constructive suggestions and have carefully responded to each of your comments point by point. In addition, we have made a comprehensive revision of the manuscript, which has significantly enhanced its overall quality. **Both the revised manuscript and updated code have been uploaded for your review.**
>
> We truly treasure the opportunity to discuss with excellent peers like you, which not only makes our work well improved, but also opens up our research horizons to better contribute to the community. As the rebuttal period is coming to an end, we welcome any further feedback you may have and would be honored to learn from you. Meantime, **we sincerely hope that you will kindly reconsider your evaluation of our work,** as it is a recognition of our work being enhanced and a great encouragement to our efforts. Thank you again for your valuable time and continued support. Wishing you all the best in your endeavors!
>
> Best regards,
>
> Authors of submission 9900

---

### Author Response · Authors · 2024-12-01
**A brief rebuttal to reviewers' comments.**

Dear Area Chair and Reviewers,

We deeply appreciate the efforts made by the Area Chair and reviewers on our manuscript. We have point-by-point responded to all the concerns raised by the reviewers and revised our manuscript accordingly.

**As the discussion period coming to an end, we would like to summarize our responses.** The major **C**oncerns of the reviewers are two folds: (**C1**) *Experiments on more tasks are required to further validate the effectiveness of the proposed COFlowNet,*  and (**C2**) *Performance of COFlowNet is slightly worse than that of QM-GFlowNet.* For your convenience, We provide a brief rebuttal below.

**(C1) Additional experiments on more tasks.** Following the suggestion of reviewers, we conducted additional experiments to evaluate COFlowNet on two more tasks: **Anti-Microbial Peptide Design** and **Item Recommendation**. The results demonstrate that COFlowNet exhibits excellent offline learning capabilities and consistent effectiveness across various tasks. We have incorporated these results and detailed analyses into our revised manuscript (**Section B**).

**(C2) Performance of COFlowNet.** COFlowNet is an offline flow-based model, and can solely learn from collected data, which covers limited state space. However, online models, including GFlowNet and QM-GFlowNet, have a better ability to explore the whole state space by interacting with the proxy model. Consequently, given a good proxy model, offline models can hardly outperform online models in generating high-score candidates. We have provided in-depth analyses in our responses and updated the manuscript (**Section 4.2.3**) to thoroughly clarify these distinctions.

We sincerely hope that our responses can address the reviewers' concerns, and warmly invite reviewers to engage in further discussions with us. Thank you once again for your valuable time and continued support. Wishing you all the best in your endeavors!

Best regards,

Authors of submission 9900

---

### Author Response · Authors · 2024-12-03
**A Gentle Reminder: Discussion Period Nearing Closure**

Dear Reviewers,

We sincerely appreciate your valued efforts and continued support of our work. **We have carefully responded to each of your comments and thoroughly revised the manuscript accordingly.** We are deeply honored that our detailed responses can address your concerns during the rebuttal process.

As the discussion period is coming to a close on **December 2nd (today)**, and If you have no further comments, **we kindly remind you to consider updating your score.**

Thank you again for your time and consideration!

Best regards,

Authors of submission 9900

---

> ### Author Response · Authors · 2024-12-03
> **Kind Reminder: Only A Few Hours Left for Discussion**
>
> Dear reviewers,
>
> We hope that everything goes well for you! During the rebuttal phase, we have done our utmost to respond to you in great detail and thoroughly revise the manuscript. We sincerely hope that our persistent efforts can satisfy you.
>
> Although there are **only a few hours left** before the end of the discussion period, **we kindly request a timely update on your evaluation of our work**. Your feedback would mean a great deal to us, and we would deeply appreciate your consideration.
>
> Thank you once again for your valuable time and continued support!
>
> Best regards,
>
> Authors of submission 9900

---

> > ### Author Response · Authors · 2024-12-03
> > **Final Follow-Up as Discussion Period Ends**
> >
> > Dear reviewers,
> >
> > We hope this message finds you well. As the discussion period is about to conclude, we would like to take this opportunity to sincerely thank you once again for your valuable time and thoughtful feedback on our work.
> >
> > During the rebuttal phase, **we have worked diligently to address your comments and thoroughly revise the manuscript, aiming to meet your expectations and clarify any concerns.** Your insights have been invaluable in improving the quality and clarity of our work, and we deeply appreciate the effort you have dedicated to reviewing our manuscript.
> >
> > While we understand the constraints of time, **we sincerely hope you might consider providing a final update to your evaluation.** Your feedback and acknowledgment of our revisions would be greatly encouraging to us as we continue our efforts to advance this work.
> >
> > Thank you again for your support and for the opportunity to learn and improve through this process. We wish you all the best in your future endeavors.
> >
> > Best regards,
> >
> > Authors of submission 9900

---

### Meta-Review · Area_Chair_q4Hz · 2024-12-15

**Metareview:**

This paper presents the Conservative Offline GFlowNet (COFlowNet) framework, aimed at enhancing the quality and diversity of generated candidates under offline setting. Overall, the paper demonstrates strong theoretical rigor, meticulous technical details, and comprehensive experimental evaluations. The majority of reviewers expressed satisfaction with the authors’ responses and provided positive feedback on the paper:

1.It introduces an offline learning framework for GFlowNets, offering a novel research perspective in this domain.

2.The method exhibits strong generalizability, achieving impressive experimental results across multiple practical tasks.

3.The comparison between offline and online methods is clear and insightful, effectively highlighting their respective application scenarios and advantages.

Thus, it is recommended that this paper be accepted.

**Additional Comments On Reviewer Discussion:**

This paper makes a strong contribution and is recommended for acceptance.

---

### Decision · Program_Chairs · 2025-01-22

Accept (Poster)